# C-Adapter: Adapting Deep Classifiers for Efficient Conformal Prediction Sets

**Kangdao Liu**[1,2,*]**, Hao Zeng**[1]**, Jianguo Huang**[3]**, Huiping Zhuang**[4]**, Chi-Man Vong**[2,†]**, Hongxin Wei**[1,†]

[1]Department of Statistics and Data Science, Southern University of Science and Technology
[2]Department of Computer and Information Science, University of Macau
[3]College of Computing and Data Science, Nanyang Technological University
[4]Shien-Ming Wu School of Intelligent Engineering, South China University of Technology

## Abstract

Conformal prediction, as an emerging uncertainty quantification technique, typically functions as post-hoc processing for the outputs of trained classifiers. To optimize the classifier for maximum predictive efficiency, Conformal Training rectifies the training objective with a regularization that minimizes the average prediction set size at a specific error rate. However, the regularization term inevitably deteriorates the classification accuracy and leads to suboptimal efficiency of conformal predictors. To address this issue, we introduce **Conformal Adapter** (C-Adapter), an adapter-based tuning method to enhance the efficiency of conformal predictors without sacrificing accuracy. In particular, we implement the adapter as a class of intra order-preserving functions and tune it with our proposed loss that maximizes the discriminability of non-conformity scores between correctly and randomly matched data-label pairs. Using C-Adapter, the model tends to produce higher non-conformity scores for incorrect labels than for correct ones, thereby enhancing predictive efficiency across different coverage rates. Extensive experiments show that C-Adapter can effectively adapt various classifiers for efficient prediction sets, as well as enhance the conformal training method.

## 1 Introduction

Quantifying the uncertainty of predictions is critical for artificial intelligence systems, particularly in high-stakes environments (e.g., financial decision-making and medical diagnostics). Conformal prediction, a statistic framework for uncertainty estimation, converts an algorithm's predictions into prediction sets containing the true class with a user-specified coverage rate (Balasubramanian et al., 2014; Shafer & Vovk, 2008). Critically, the validity of sets is satisfied in a distribution-free sense: they possess explicit, non-asymptotic guarantees even without distributional assumptions or model assumptions. To obtain informative outputs, it is of great importance to improve the *efficiency* of conformal predictors, aiming for the prediction sets with minimal ambiguity (Sadinle et al., 2019).

Conformal prediction typically functions as post-hoc processing for the output of trained classifiers, which might already be either unnecessarily conservative or overconfident (Bellotti, 2021; Stutz et al., 2022). To optimize the predictive efficiency, Conformal Training (Stutz et al., 2022) rectifies the training objective with a regularization that minimizes the average prediction set size at a specific error rate (e.g., 0.01). However, the regularization term inevitably deteriorates the classifier accuracy by increasing the difficulty of converging to an optimal solution (Stutz et al., 2022), which in turn leads to the suboptimal efficiency of the conformal predictor. This challenge is especially significant when dealing with many classes, making it difficult to apply to large-scale datasets such as ImageNet (Deng et al., 2009). This motivates our methodology, which enables the efficient adaptation of trained classifiers for conformal prediction without sacrificing classification accuracy.

In this work, we propose *Conformal Adapter* (dubbed **C-Adapter**), an adapter-based tuning method to enhance the efficiency of conformal predictors. In particular, we tune an adapter layer appended

---

*Work was done as a research intern at Southern University of Science and Technology.
†Correspond to `weihx@sustech.edu.cn`, `cmvong@um.edu.mo`.

to trained classifiers for conformal prediction using the training data. Our key idea is to adapt trained classifiers for conformal prediction while preserving the ranking of labels in the output logits, thereby maintaining the top-$k$ accuracy of the classifiers. To achieve this, we implement the adapter as a class of intra order-preserving functions (Rahimi et al., 2020). For the optimization of this adapter, we propose a loss function that enhances the discriminability of non-conformity scores between correctly and randomly matched data-label pairs. In effect, the loss encourages the non-conformity scores of correctly matched data-label pairs to be lower than those of incorrectly matched ones, resulting in more efficient predictions across different coverage rates. Equipped with C-Adapter, the predictor maintains top-$k$ accuracy and generates highly efficient prediction sets. For better clarity, we include a diagram in Appendix B to visually illustrate the application of C-Adapter.

To validate our method, we conduct extensive evaluations on three benchmarks of image classification, including CIFAR-100 (Krizhevsky et al., 2009), ImageNet (Deng et al., 2009), and ImageNet-V2 (Recht et al., 2019). The results demonstrate that C-Adapter can significantly enhance the efficiency of conformal predictors. For example, C-Adapter reduces the average size for APS from 9.21 to 2.86 on ImageNet (Deng et al., 2009) with DenseNet121 (Huang et al., 2017) at $\alpha = 0.1$. This approach also generalizes effectively to different score functions, consistently improving their efficiency. Moreover, we show that C-Adapter can improve the efficiency of prediction sets while either enhancing or maintaining conditional coverage metrics. Notably, our method is easy to implement, as it does not require heavy tuning of hyperparameters and incurs low computational costs.

We summarize our contributions as follows:

- We propose C-Adapter, a simple and effective method to enhance the efficiency of conformal predictors without sacrificing classifier accuracy. This approach serves as a distinctive complement to existing score-based and training-based conformal prediction algorithms.

- We theoretically demonstrate that enhancing the discriminability of non-conformity scores between correctly and randomly matched data-label pairs is equivalent to improving the overall efficiency of conformal predictors. To this end, we propose a loss function specifically designed to achieve this goal and apply it to optimize our conformal adapter.

- We empirically show that C-Adapter effectively adapts a range of classifiers for efficient prediction sets across different score functions. Moreover, we show that C-Adapter outperforms the fine-tuned version of Conformal Training and further improves its performance.

## 2 BACKGROUND

**Setup** In this work, we consider the multi-class classification task with $K$ classes. Let $(X, Y) \sim \mathcal{P}_{\mathcal{X}\mathcal{Y}}$ denote a random data pair sampled from the joint distribution $\mathcal{P}_{\mathcal{X}\mathcal{Y}}$, where $\mathcal{X} \subset \mathbb{R}^d$ is the input space and $\mathcal{Y} := \{1, \cdots, K\}$ is the label space. Given a training set, we learn a classifier $f : \mathcal{X} \to \mathbb{R}^K$ with parameter $\boldsymbol{\theta}$. Given an instance $\boldsymbol{x}$, we predict the probability of class $k$ by:

$$\hat{\pi}_k(\boldsymbol{x}; \boldsymbol{\theta}) = \psi(f_k(\boldsymbol{x}; \boldsymbol{\theta})) = \frac{e^{f_k(\boldsymbol{x}; \boldsymbol{\theta})}}{\sum_{i=1}^{K} e^{f_i(\boldsymbol{x}; \boldsymbol{\theta})}}, \tag{1}$$

where $\psi$ denotes the softmax function and $f_k(\boldsymbol{x}; \boldsymbol{\theta})$ is the $k$-th element of the logits $f(\boldsymbol{x}; \boldsymbol{\theta})$. Deep classifiers always suffer from the miscalibration issue: the estimated probabilities might be either conservative or overconfident, leading to inaccurate assessments of uncertainty (Guo et al., 2017).

**Conformal Prediction** In uncertainty quantification, conformal prediction (Vovk et al., 2005) seeks to construct prediction sets $\mathcal{C}(X) \subseteq \mathcal{Y}$ such that $\mathbb{P}\{Y \in \mathcal{C}(X)\} \geq 1 - \alpha$ for a pre-specified error rate $\alpha \in (0, 1)$. To satisfy the desired coverage rate $1 - \alpha$, we take an independent conformal calibration dataset $\mathcal{D}_{\mathrm{cal}} := \{(\boldsymbol{x}_i, y_i)\}_{i=1}^n$, and then determine the threshold $\tau_\alpha$ such that the prediction sets are large enough to achieve the desired coverage level of $1 - \alpha$ on this calibration set. Specifically, we calculate the non-conformity score $s_i := S(\boldsymbol{x}_i, y_i; \hat{\pi})$ for each sample $(\boldsymbol{x}_i, y_i)$ in the calibration set where $S$ is a pre-specified score function to measure non-conformity of each input sample. We then determine the threshold $\tau_\alpha$ as the $1 - \alpha$ quantile of the set $\{s_i\}_{i=1}^n$, as follows:

$$\tau_\alpha = \inf \left\{ s : \frac{|\{i \in \{1, \cdots, n\} : s_i \leq s\}|}{n} \geq \frac{\lceil (n+1)(1-\alpha) \rceil}{n} \right\}.$$

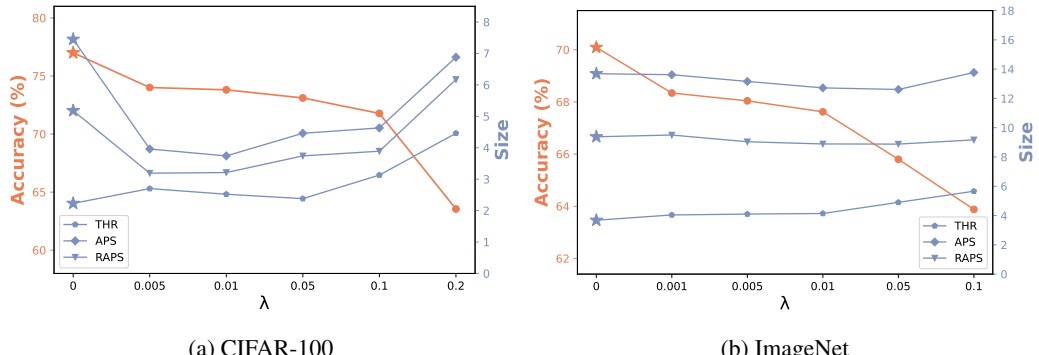

(a) CIFAR-100
(b) ImageNet

Figure 1: **The accuracy and efficiency of ConfTr with various $\lambda$**, using THR, APS and RAPS at $\alpha = 0.1$. The experiments are conducted with ResNet18 on (a) CIFAR-100 and (b) ImageNet. ★ represents the baseline without ConfTr. The findings indicate that the increment of $\lambda$ decreases the classification accuracy, ultimately leading to suboptimal efficiency in conformal prediction.

During testing, we calculate the non-conformity score $S(\boldsymbol{x}_{n+1}, y; \hat{\pi})$ for a given instance $\boldsymbol{x}_{n+1}$ and each label $y \in \mathcal{Y}$. Then, the prediction set $\mathcal{C}(\boldsymbol{x}_{n+1}; \tau_\alpha, \hat{\pi})$ with $1 - \alpha$ coverage is constructed by:

$$\mathcal{C}(\boldsymbol{x}_{n+1}; \tau_\alpha, \hat{\pi}) := \{y \in \mathcal{Y} : S(\boldsymbol{x}_{n+1}, y; \hat{\pi}) \le \tau_\alpha\}. \tag{2}$$

In other words, the final prediction sets achieve *marginal coverage* by containing all labels with non-conformity scores below the threshold (Vovk, 2012; Angelopoulos et al., 2020). In addition to the coverage, we typically expect to optimize the average size of prediction sets, which is referred to as *efficiency*. Nevertheless, the length of the resulting prediction sets can vary dramatically depending on the design of $S(\boldsymbol{x}, y; \hat{\pi})$. In this work, we consider three popular score functions for classification, including THR (Sadinle et al., 2019), APS (Romano et al., 2020), and RAPS (Angelopoulos et al., 2020). We provide a detailed introduction to these score functions in Appendix C.1.

**Conformal Training** Conformal prediction typically works as post-hoc processing for the outputs of trained classifiers. To optimize the classifier for maximum predictive efficiency, Conformal Training (ConfTr) (Stutz et al., 2022) rectifies the training objective with a regularization that minimizes the average prediction set size at a specific error rate $\alpha$. The loss function is formulated as:

$$\mathcal{L}_{\text{ConfTr}}(f(\boldsymbol{x}; \boldsymbol{\theta}), y, \tau_\alpha^{\text{soft}}) = \mathcal{L}_{\text{cls}}(f(\boldsymbol{x}; \boldsymbol{\theta}), y) + \lambda \mathcal{L}_{\text{size}}(f(\boldsymbol{x}; \boldsymbol{\theta}), \tau_\alpha^{\text{soft}}). \tag{3}$$

Here, $\mathcal{L}_{\text{cls}}$ represents the classification loss, while $\mathcal{L}_{\text{size}}$ refers to the size loss, which approximates the size of the prediction set at a coverage rate of $1 - \alpha$. $\tau_\alpha^{\text{soft}}$ denotes the soft threshold and the hyperparameter $\lambda$ controls the strength of the regularization term. We provide a detailed introduction to ConfTr in Appendix C.2. Notably, while ConfTr with a tuned hyperparameter $\lambda$ may improve the efficiency of conformal predictors, the regularization term $\mathcal{L}_{\text{size}}$ inevitably deteriorates the classification accuracy of the classifier by increasing the difficulty of converging to an optimal solution (Stutz et al., 2022). We theoretically demonstrate in Appendix D that the lower bound on the expected size of the conformal prediction set is inversely related to the top-$k$ accuracy of the base classifier. Similar analyses exploring the relationship between model performance and the efficiency of conformal predictors have been presented in prior works (Zecchin et al., 2024; Sadinle et al., 2019).

To provide a straightforward view, we demonstrate the effect of the regularization term $\mathcal{L}_{\text{size}}$ on the accuracy and efficiency of conformal predictors in Figure 1. We conduct experiments of ConfTr with various $\lambda$, using ResNet18 (He et al., 2016) on CIFAR100 and ImagNet. The results demonstrate that using this regularization continuously degrades the classification accuracy of the classifier as $\lambda$ increases. For efficiency, ConfTr raises the average size of APS and RAPS after achieving the optimal performance on CIFAR-100. On ImageNet, ConfTr offers only marginal benefits for the efficiency of conformal predictors. The negative effect of ConfTr is especially noticeable on THR: the average size of THR is consistently increased over various $\lambda$. The decrease in classification accuracy inevitably results in larger prediction sets, which in turn limits the efficiency on average. We present a detailed description of the experimental setup and the effect of the regularization term $\mathcal{L}_{\text{size}}$ on top-$k$ accuracy in Appendix I.1. We proceed by introducing our method, targeting this issue.

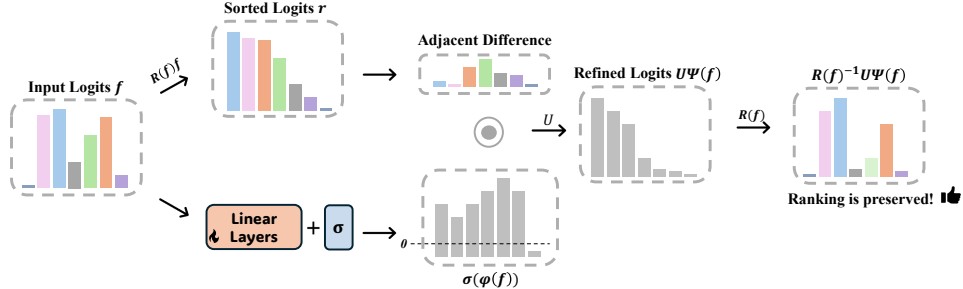

Figure 2: **Flow of C-Adapter.** The design follows the definition of intra order-preserving functions (Rahimi et al., 2020), ensuring that the refined logits maintain the ranking of the inputs.

## 3 METHOD

In our previous analysis, we demonstrate that ConfTr deteriorates the classification accuracy, thereby hindering the efficiency of conformal predictors. To address this issue, our key idea is to adapt the trained classifiers for conformal prediction while preserving the ranking of labels in the output logits, thereby keeping the top-$k$ accuracy of the original classifier unchanged.

**Conformal Adapter**    To this end, we propose a novel adapter-based tuning method – *Conformal Adapter* (dubbed **C-Adapter**), which appends an adapter layer to trained classifiers for conformal prediction. Formally, we use $g : \mathbb{R}^K \to \mathbb{R}^K$ to denote the conformal adapter that takes the model outputs $f(\boldsymbol{x}; \boldsymbol{\theta})$ as input. Then, the final prediction of the model equipped with C-Adapter is:

$$\widetilde{\pi}(\boldsymbol{x}; \boldsymbol{\theta}, \boldsymbol{w}) = \psi(g(f(\boldsymbol{x}; \boldsymbol{\theta}); \boldsymbol{w})),$$

where $\boldsymbol{w}$ denotes the parameters of C-Adapter. While ConfTr alters the parameters of trained classifiers $\boldsymbol{\theta}$ through retraining or fine-tuning, we only update a few trainable parameters $\boldsymbol{w}$ added for conformal prediction. In addition to enhancing training efficiency, the adapter-based tuning method requires access only to the model outputs. This makes it compatible with black-box models (e.g., online APIs) and other modern neural networks (e.g., Radford et al. (2021, CLIP)).

Importantly, the adapter requires to be learned within a hypothesis space that can provably guarantee preserving the accuracy of the original network $f$. To achieve that, we implement the adapter as a class of *intra order-preserving functions* (Rahimi et al., 2020), a family of functions that is both necessary and sufficient to keep the top-$k$ accuracy of the original network unchanged. Formally, a function $h : \mathbb{R}^K \to \mathbb{R}^K$ is *intra order-preserving*, if, for all $i, j \in [K]$ and any vector $\boldsymbol{x} \in \mathbb{R}^K$, $\boldsymbol{x}_i > \boldsymbol{x}_j$ (or $\boldsymbol{x}_i = \boldsymbol{x}_j$) if and only if $h_i(\boldsymbol{x}) > h_j(\boldsymbol{x})$ (or $h_i(\boldsymbol{x}) = h_j(\boldsymbol{x})$). For convenience, we use $\boldsymbol{f}$ to indicate the model output $f(\boldsymbol{x}; \boldsymbol{\theta})$. We denote $R : \mathbb{R}^K \to \mathbb{U}^K$ as the sorting function, where $\mathbb{U}^K \subset \{0, 1\}^{K \times K}$ represents the set of $K \times K$ permutation matrices. We have $\boldsymbol{r} = R(\boldsymbol{f})\boldsymbol{f}$ as the sorted $\boldsymbol{f}$, satisfying $\boldsymbol{r}_1 > \cdots > \boldsymbol{r}_K$. We use $U$ to denote the $K \times K$ upper-triangular matrix of ones.

To ensure that C-Adapter belongs to the class of *intra order-preserving functions*, we define it by

$$g(\boldsymbol{f}; \boldsymbol{w}) = R(\boldsymbol{f})^{-1} U \Psi(\boldsymbol{f}), \tag{4}$$

where the $i$-th term of $\Psi(\boldsymbol{f})$ is formulated as:

$$\Psi_i(\boldsymbol{f}) = \begin{cases} \sqrt{(\boldsymbol{r}_i - \boldsymbol{r}_{i+1})}\sigma(\varphi_i(\boldsymbol{f})) & \text{for } i < K, \\ \varphi_K(\boldsymbol{f}) & \text{for } i = K. \end{cases} \tag{5}$$

Here, $\varphi(\boldsymbol{f}) = \boldsymbol{w} \cdot \boldsymbol{f} + \boldsymbol{w}'$, and $\sigma$ represents the sigmoid function. We denote $\varphi_i(\boldsymbol{f})$ as the $i$-th component of $\varphi(\boldsymbol{f})$. The term $\sqrt{\boldsymbol{r}_i - \boldsymbol{r}_{i+1}}$ is designed to preserve all ties and inequalities in the sorted sequence, and equals zero if and only if $\boldsymbol{r}_i = \boldsymbol{r}_{i+1}$. Alternative formulations, such as $(\boldsymbol{r}_i - \boldsymbol{r}_{i+1})$ or $\mathbb{1}_{\{\boldsymbol{r}_i > \boldsymbol{r}_{i+1}\}}$, are also valid options within our framework, each resulting in different local optima for this task. We outline the workflow in Figure 2. A detailed description of this function family, along with the configuration of our adapter and optimization strategies, is provided in Appendix E. The core idea is to decouple the label ranking and the logit values in the tuning. It begins by preserving a duplicate of the label ranking, and then transmit the logits to the linear layer for processing. Finally, we recover the label ranking in the output. This structure decouples the logit order from the adaptation for conformal prediction, allowing C-Adapter to focus on optimizing efficiency. We show the superiority of this adaptation strategy over others in Figures 5 and 6.

**Training objective** ConfTr optimizes the efficiency of conformal predictors at a predetermined error rate (e.g., $\alpha = 0.01$), with the goal of enabling the predictor to generalize across all error rates. In contrast, we consider a more general criterion for predictive efficiency:

$$\mathbb{E}_{\boldsymbol{x} \sim \mathcal{P}_{\mathcal{X}}} \left[ \int_0^1 |\mathcal{C}(\boldsymbol{x}; \tau_\alpha, \widetilde{\pi}_{\boldsymbol{w}})| \, \mathrm{d}\alpha \right], \tag{6}$$

which measures the definite integral of efficiency over $\alpha \in (0, 1)$. For notation shorthand, we use $\widetilde{\pi}_{\boldsymbol{w}}$ to indicate that the underlying classifier $f$ is equipped with C-Adapter, parameterized by $\boldsymbol{w}$. This objective is analogous to the AUC in classification (Cortes & Mohri, 2003), as AUC reflects the classifier's performance across all possible thresholds, while classification error considers only a single fixed one. However, the objective in Equation (6) cannot be directly computed from a given dataset. To address this issue, we translate it into an equivalent form that can be explicitly calculated.

From Equation (2), we can infer that we construct the conformal prediction set for $\hat{X} \sim \mathcal{P}_{\mathcal{X}}$ at $\alpha$ by comparing the non-conformity score $S(\hat{X}, y; \widetilde{\pi}_{\boldsymbol{w}})$ with $\tau_\alpha$ for each $y \in \mathcal{Y}$. Therefore, it is straightforward to verify that the expected set size at the error rate $\alpha$ over the data distribution $\mathcal{P}_{\mathcal{X}}$ is determined by the probability of the event $\{\tau_\alpha \geq S(\hat{X}, \hat{Y}; \widetilde{\pi}_{\boldsymbol{w}})\}$, where $\hat{X} \sim \mathcal{P}_{\mathcal{X}}$ and $\hat{Y} \sim \text{Uniform}(\mathcal{Y})$. When extending to any $\alpha \in (0, 1)$, the threshold $\tau_\alpha$ can be the non-conformity score of any observation $(X, Y) \sim \mathcal{P}_{\mathcal{X}\mathcal{Y}}$. This prompts us to consider the following probability:

$$\mathbb{P}\left( S(X, Y; \widetilde{\pi}_{\boldsymbol{w}}) \geq S(\hat{X}, \hat{Y}; \widetilde{\pi}_{\boldsymbol{w}}) \right), \text{where } (X, Y) \sim \mathcal{P}_{\mathcal{X}\mathcal{Y}}, \hat{X} \sim \mathcal{P}_{\mathcal{X}}, \hat{Y} \sim \text{Uniform}(\mathcal{Y}). \tag{7}$$

In particular, this probability quantifies the likelihood that the non-conformity score of a randomly matched data-label pair $(\hat{X}, \hat{Y})$ is not greater than that of a correctly matched pair $(X, Y)$. This probability approaches zero when the scores of correctly and incorrectly matched data-label pairs are well distinguishable, and approaches $1/2$ when they are not effectively distinguished. In the following, we present a formal analysis demonstrating that minimizing the probability in Equation (7) is equivalent to optimizing the overall efficiency defined in Equation (6).

**Proposition 1.** *Let $\hat{\pi}$ and $\hat{\pi}'$ be pre-trained classifiers with parameters $\theta$ and $\theta'$, respectively, and let $S$ be a specific non-conformity score function. We denote $\mathcal{P}_{S_\theta}$ and $\mathcal{P}_{S_{\theta'}}$ as the distributions of $S(X, Y; \hat{\pi})$ and $S(X, Y; \hat{\pi}')$, where $(X, Y) \sim \mathcal{P}_{\mathcal{X}\mathcal{Y}}$. Let $F_{S_\theta}$, and $F_{S_{\theta'}}$ be the CDF corresponding to $\mathcal{P}_{S_\theta}$ and $\mathcal{P}_{S_{\theta'}}$. Given that $\hat{X} \sim \mathcal{P}_{\mathcal{X}}$ and $\hat{Y}$ follows a uniform distribution over $\mathcal{Y}$, we have*

$$\mathbb{P}\left( S(X, Y; \hat{\pi}) \geq S(\hat{X}, \hat{Y}; \hat{\pi}) \right) > \mathbb{P}\left( S(X, Y; \hat{\pi}') \geq S(\hat{X}, \hat{Y}; \hat{\pi}') \right)$$

*holds if and only if*

$$\mathbb{E}_{X \sim \mathcal{P}_{\mathcal{X}}} \left[ \int_0^1 |\mathcal{C}\left(X; F_{S_\theta}^{-1}(1-\alpha), \hat{\pi}\right)| \, \mathrm{d}\alpha \right] > \mathbb{E}_{X \sim \mathcal{P}_{\mathcal{X}}} \left[ \int_0^1 |\mathcal{C}\left(X; F_{S_{\theta'}}^{-1}(1-\alpha), \hat{\pi}'\right)| \, \mathrm{d}\alpha \right].$$

The proof of Proposition 1 is provided in Appendix F. Here, the inverse CDF calculates the $(1-\alpha)$-th quantile of the score distributions, which determines the threshold $\tau_\alpha$. Then, to optimize overall efficiency in Equation (6), we turn to minimize the following objective, rewritten from Equation (7):

$$\mathcal{L}(\boldsymbol{w}) = \mathbb{E}\left[ \mathbb{1}_{\{S(X,Y;\widetilde{\pi}_{\boldsymbol{w}}) > S(\hat{X}, \hat{Y}; \widetilde{\pi}_{\boldsymbol{w}})\}} \right], \tag{8}$$

where $(X, Y) \sim \mathcal{P}_{\mathcal{X}\mathcal{Y}}, \hat{X} \sim \mathcal{P}_{\mathcal{X}}$, and $\hat{Y} \sim \text{Uniform}(\mathcal{Y})$. Given the non-differentiability of the indicator function, it is common practice to utilize surrogate functions as differentiable approximations (Yan et al., 2003; Yuan et al., 2021). In this work, we apply the sigmoid function with a parameter $T$ as the surrogate, defined as $\sigma_T(x) = 1/(1 + \exp(-x/T))$. For the score function utilized during training, we employ either THR or APS. The differentiable APS is implemented as outlined in ConfTr (Stutz et al., 2022). Ultimately, the convex relaxation of Equation (8) is given by

$$\widetilde{\mathcal{L}}(\boldsymbol{w}) = \mathbb{E}\left[ \sigma_T\left( S(X, Y; \widetilde{\pi}_{\boldsymbol{w}}) - S(\hat{X}, \hat{Y}; \widetilde{\pi}_{\boldsymbol{w}}) \right) \right]. \tag{9}$$

By optimizing this objective, the scores of correctly and incorrectly matched data-label pairs become more distinguishable: correctly matched pairs are encouraged to have relatively smaller non-conformity scores compared to incorrectly matched pairs. We visualize this effect in Figure 3. With C-Adapter, the APS scores of incorrect labels become significantly higher than those of correct labels, leading to more efficient prediction sets across varying coverage rates.

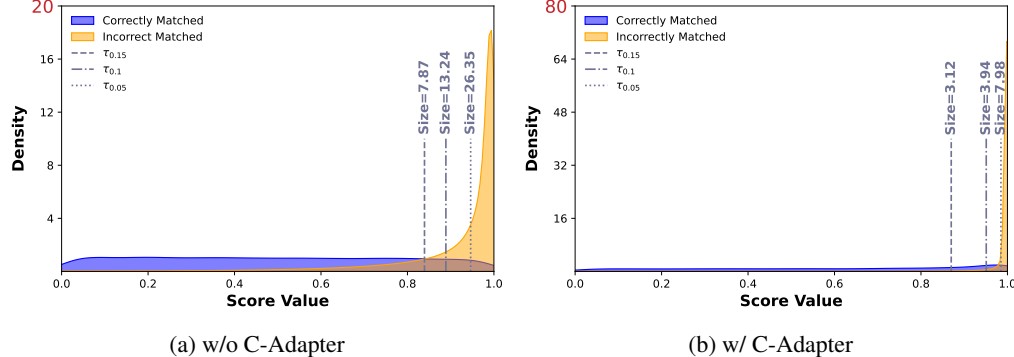

(a) w/o C-Adapter           (b) w/ C-Adapter

Figure 3: **Score distributions of correctly and incorrectly matched data-label pairs:** (a) without C-Adapter, (b) with C-Adapter. We calculate the APS scores on ImageNet using CLIP (Radford et al., 2021). The three gray lines indicate the set sizes with $\tau_\alpha$ at $\alpha = 0.15, 0.1,$ and $0.05$, respectively. Using C-Adapter, the APS scores of incorrect labels tend to be much higher (approaching the maximum 1.0) than those of correct labels. The highly distinguishable scores between correct and incorrect labels translate to more efficient conformal prediction sets at different coverage rates.

**Batched optimization** In the $t$-th iteration, we construct an auxiliary batch $\hat{\mathcal{B}}_t$ by creating $K$ data-label pairs for each instance in $\mathcal{B}_t$. Each pair $(\hat{\boldsymbol{x}}, \hat{y})$ in $\hat{\mathcal{B}}_t$ consists of an instance from $\mathcal{B}_t$ and one of the $K$ possible labels $\hat{y} \in \mathcal{Y}$. Subsequently, we update the parameters $\boldsymbol{w}$ of C-Adapter by

$$\boldsymbol{w}^{(t)} \leftarrow \boldsymbol{w}^{(t-1)} - \eta_t \cdot \nabla_{\boldsymbol{w}} \left[ \frac{1}{|\mathcal{B}_t| \cdot |\hat{\mathcal{B}}_t|} \sum_{(\boldsymbol{x}, y) \in \mathcal{B}_t} \sum_{(\hat{\boldsymbol{x}}, \hat{y}) \in \hat{\mathcal{B}}_t} \sigma_T \left( S\left(\boldsymbol{x}, y; \widetilde{\pi}_{\boldsymbol{w}}\right) - S\left(\hat{\boldsymbol{x}}, \hat{y}; \widetilde{\pi}_{\boldsymbol{w}}\right) \right) \right]. \quad (10)$$

The optimization incurs low computational costs, as we only update the parameters of the linear layers in C-Adapter. In practical applications, we tune the parameters of C-Adapter using the training set for the trained classifier $f$. Our method can also be implemented with a hold-out set, which is explicitly validated in Figure 8. Noticeably, our method offers several compelling advantages:

- **Flexible:** C-Adapter can enhance the efficiency of conformal predictors across different non-conformity score functions, not limited to the one employed during its tuning (see Table 1 and Table 5). By default, we tune C-Adapter using THR.

- **Easy to use:** C-Adapter requires minimal hyperparameter tuning and performs well with any sufficiently small $T$ (see Figure 7). Moreover, our method shows high computational efficiency and a rapid convergence rate (refer to the convergence analysis in Appendix G).

- **Model-agnostic:** C-Adapter requires access only to the model outputs and integrates effortlessly with any classifier. Our method can effectively adapt trained classifiers for efficient prediction sets, regardless of the network architecture or pre-training strategy.

## 4 EXPERIMENTS

### 4.1 EXPERIMENTAL SETUP

**Dataset** We evaluate our approach using three benchmarks of image classification: CIFAR-100 (Krizhevsky et al., 2009), ImageNet (Deng et al., 2009), and ImageNet-V2 (Recht et al., 2019). For CIFAR-100 and ImageNet-V2, we randomly split the test sets into calibration and test subsets, each containing 5,000 samples. For ImageNet, we partition the 50,000-sample test set into a calibration subset of 30,000 samples and a test subset of 20,000 samples.

**Models** For our evaluations, we utilize four well-established deep image classifiers: ResNet101 (RN101) (He et al., 2016), two variants of DenseNet (DN121 and DN161) (Huang et al., 2017), and ResNeXt50 (RNX50) (Xie et al., 2017). Additionally, we employ the Vision-Language Model CLIP (Radford et al., 2021), which is based on a Vision Transformer architecture (ViT-B/16) (Dosovitskiy et al., 2020). For ImageNet, we leverage pre-trained classifiers from TorchVision (Paszke et al.,

Table 1: **Performance of C-Adapter on common benchmarks.** ↓ indicates that a smaller value is better. Results in **bold** indicate superior performance. C-Adapter is tuned using THR.

| Score | Model | w/o C-Adapter \ w/ C-Adapter | | | | | | | |
|---|---|---|---|---|---|---|---|---|---|
| | | **ImageNet** | | | | **CIFAR-100** | | | |
| | | $\alpha = 0.05$ | | $\alpha = 0.1$ | | $\alpha = 0.05$ | | $\alpha = 0.1$ | |
| | | Coverage | Size ($\downarrow$) | Coverage | Size ($\downarrow$) | Cover | Size ($\downarrow$) | Coverage | Size ($\downarrow$) |
| THR | RN101 | 0.95 \ 0.95 | 4.03 \ **3.82** | 0.90 \ 0.90 | 1.91 \ **1.89** | 0.95 \ 0.95 | 3.64 \ **3.17** | 0.90 \ 0.90 | 1.87 \ **1.76** |
| | DN121 | 0.95 \ 0.95 | 5.66 \ **5.35** | 0.90 \ 0.90 | 2.42 \ **2.34** | 0.95 \ 0.95 | 3.27 \ **3.00** | 0.90 \ 0.90 | 1.72 \ **1.70** |
| | DN161 | 0.95 \ 0.95 | 4.03 \ **3.69** | 0.90 \ 0.90 | 1.89 \ **1.82** | 0.95 \ 0.95 | 2.91 \ **2.75** | 0.90 \ 0.90 | 1.72 \ **1.69** |
| | RNX50 | 0.95 \ 0.95 | 4.06 \ **3.87** | 0.90 \ 0.90 | 1.87 \ **1.85** | 0.95 \ 0.95 | 3.41 \ **3.09** | 0.90 \ 0.90 | 1.78 \ **1.76** |
| | CLIP | 0.95 \ 0.95 | 6.88 \ **6.71** | 0.90 \ 0.90 | 3.33 \ **3.25** | 0.95 \ 0.95 | 9.71 \ **8.25** | 0.90 \ 0.90 | 4.78 \ **4.36** |
| | **Average** | 0.95 \ 0.95 | 4.93 \ **4.69** | 0.90 \ 0.90 | 2.29 \ **2.23** | 0.95 \ 0.95 | 4.59 \ **4.05** | 0.90 \ 0.90 | 2.37 \ **2.25** |
| APS | RN101 | 0.95 \ 0.95 | 14.73 \ **3.98** | 0.90 \ 0.90 | 7.23 \ **2.30** | 0.95 \ 0.95 | 7.60 \ **3.19** | 0.90 \ 0.90 | 3.95 \ **1.86** |
| | DN121 | 0.95 \ 0.95 | 20.00 \ **5.73** | 0.90 \ 0.90 | 9.21 \ **2.86** | 0.95 \ 0.95 | 10.20 \ **3.08** | 0.90 \ 0.90 | 5.39 \ **1.85** |
| | DN161 | 0.95 \ 0.95 | 16.43 \ **4.23** | 0.90 \ 0.90 | 6.82 \ **2.33** | 0.95 \ 0.95 | 9.90 \ **2.86** | 0.90 \ 0.90 | 5.42 \ **1.80** |
| | RNX50 | 0.95 \ 0.95 | 21.54 \ **4.26** | 0.90 \ 0.90 | 8.92 \ **2.32** | 0.95 \ 0.95 | 9.95 \ **3.26** | 0.90 \ 0.90 | 5.14 \ **1.91** |
| | CLIP | 0.95 \ 0.95 | 26.35 \ **7.98** | 0.90 \ 0.90 | 13.24 \ **3.94** | 0.95 \ 0.95 | 16.13 \ **13.50** | 0.90 \ 0.90 | 10.18 \ **8.70** |
| | **Average** | 0.95 \ 0.95 | 19.81 \ **5.24** | 0.90 \ 0.90 | 9.08 \ **2.75** | 0.95 \ 0.95 | 10.76 \ **5.18** | 0.90 \ 0.90 | 6.01 \ **3.22** |
| RAPS | RN101 | 0.95 \ 0.95 | 7.13 \ **3.75** | 0.90 \ 0.90 | 4.60 \ **2.25** | 0.95 \ 0.95 | 5.16 \ **4.43** | 0.90 \ 0.90 | 3.25 \ **1.81** |
| | DN121 | 0.95 \ 0.95 | 10.28 \ **6.53** | 0.90 \ 0.90 | 6.57 \ **2.80** | 0.95 \ 0.95 | 7.19 \ **3.74** | 0.90 \ 0.90 | 4.50 \ **1.80** |
| | DN161 | 0.95 \ 0.95 | 7.31 \ **4.10** | 0.90 \ 0.90 | 4.63 \ **2.27** | 0.95 \ 0.95 | 7.10 \ **3.15** | 0.90 \ 0.90 | 4.59 \ **1.79** |
| | RNX50 | 0.95 \ 0.95 | 7.87 \ **4.11** | 0.90 \ 0.90 | 5.20 \ **2.26** | 0.95 \ 0.95 | 7.20 \ **3.94** | 0.90 \ 0.90 | 4.47 \ **1.89** |
| | CLIP | 0.95 \ 0.95 | 15.14 \ **7.82** | 0.90 \ 0.90 | 9.25 \ **3.49** | 0.95 \ 0.95 | 14.52 \ **11.19** | 0.90 \ 0.90 | 9.41 \ **7.62** |
| | **Average** | 0.95 \ 0.95 | 9.55 \ **5.26** | 0.90 \ 0.90 | 6.05 \ **2.61** | 0.95 \ 0.95 | 8.24 \ **5.45** | 0.90 \ 0.90 | 5.24 \ **2.98** |

(a) THR       (b) APS       (c) RAPS

Figure 4: **Comparison of C-Adapter and ConfTr,** using (a) THR, (b) APS, and (c) RAPS at $\alpha = 0.1$ on CIFAR-100. "ConfTr + Ours" refers to applying C-Adapter to models that have been fine-tuned using ConfTr. The results demonstrate that C-Adapter outperforms ConfTr.

2019), whereas for CIFAR-100, we train the classifiers from scratch using the entire training set. For CLIP, we rely on its inherent zero-shot capabilities to perform classification tasks.

**Implementation details** C-Adapter is tuned using Adam (Kingma & Ba, 2014) optimizer, with a batch size of 256 and a learning rate of 0.1. The parameter $T$ is set to 0.0001 by default. We partition the calibration set into a validation subset and a calibration subset in an 20:80 ratio, with the validation set used for early stopping. When a validation set is not necessary, the entire calibration set is employed for calibration, ensuring all methods have access to the same dataset. We consider three score functions: THR, APS, and RAPS. For RAPS, we set $k_{reg} = 1$ and $\lambda = 0.001$. Each experiment is repeated 10 times with different seeds, and the average result is reported. All the experiments are conducted on an NVIDIA GeForce RTX 4090 using PyTorch (Paszke et al., 2019).

**Evaluation metrics** The primary metrics for evaluating prediction sets are: (1) efficiency (Size) and (2) marginal coverage rate (Coverage). We detail these metrics in Appendix H. Moreover, we examine the impact of C-Adapter on conditional coverage metrics in Appendix J.

## 4.2 RESULTS

**C-Adapter improves the efficiency of conformal predictors.** In Table 1, we present the performance of THR, APS, and RAPS with C-Adapter on ImageNet and CIFAR-100. A salient observation is that our method drastically improves the efficiency of conformal predictors with the desired

Table 2: **Comparison of C-Adapter with different loss functions,** on ImageNet with DN121. Baseline represents the scenario without C-Adapter. Since each entry achieves the desired coverage, only **Size** is presented. Our loss achieves superior average performance compared to the size loss.

| | THR | | | | | | | APS | | | | | | |
|---|---|---|---|---|---|---|---|---|---|---|---|---|---|---|
| $\alpha$ | 0.06 | 0.05 | 0.04 | 0.03 | 0.02 | 0.01 | Average | 0.06 | 0.05 | 0.04 | 0.03 | 0.02 | 0.01 | Average |
| Baseline | 4.35 | 5.66 | 7.26 | 10.46 | 15.91 | 33.84 | 12.91 | 15.94 | 20.00 | 24.42 | 32.62 | 48.13 | 91.49 | 38.77 |
| size loss | **4.26** | **5.33** | 7.04 | 9.93 | 17.44 | 43.16 | 14.53 | 4.48 | **5.71** | 7.39 | 10.82 | 18.82 | 42.63 | 14.98 |
| Ours | 4.27 | 5.35 | **6.94** | **9.75** | **15.01** | **30.31** | **11.94** | 4.44 | 5.73 | **7.37** | **10.70** | **17.30** | **36.24** | **13.63** |

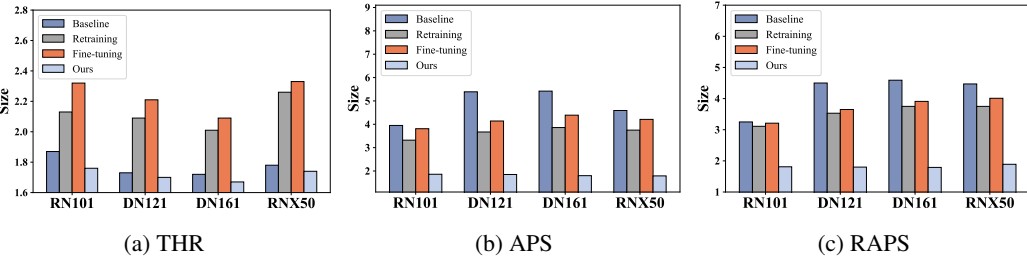

(a) THR          (b) APS          (c) RAPS

Figure 5: **Comparison of different adaptation strategies,** using (a) THR, (b) APS, and (c) RAPS at $\alpha = 0.1$. The experiment is conducted on CIFAR-100. *Retraining* refers to training the classifier from scratch with our proposed loss function, while *Fine-tuning* indicates tuning only the fully connected layer with our loss. C-Adapter outperforms the other two adaptation strategies.

coverage rate. For example, C-Adapter reduces the size of APS from 16.43 to 4.23 on ImageNet using DN161 with $\alpha = 0.05$. Notably, the improvements remain substantial when there is a mismatch between the score functions used during adapter tuning (THR) and those employed in conformal prediction (APS and RAPS). When C-Adapter is tuned with APS, similar enhancements are observed with both THR and RAPS, as detailed in Appendix K. This highlights the flexibility of our method. Overall, empirical results show that C-Adapter can enhance the efficiency of conformal predictors across various score functions, regardless of model architectures and pre-training strategies.

**C-Adapter outperforms the fine-tuned version of ConfTr.** ConfTr (Stutz et al., 2022) can also be employed as a fine-tuning method to adapt classifiers for conformal prediction. Initially, the classifier is trained with cross-entropy loss, and then only the linear layer is tuned using the objective in Equation (3). We compare this approach with ours on CIFAR-100. For ConfTr, we set the learning rate to 0.001 with a batch size of 256. A higher learning rate significantly decreases accuracy, leading to a dramatic decline in efficiency. The parameters $T$ and $\lambda$ are tuned from the sets $\{0.01, 0.1, 0.5, 1\}$ and $\{0.005, 0.01, 0.05, 0.1, 0.2\}$, respectively. During training, we utilize THRLP (Stutz et al., 2022) for ConfTr, setting $\alpha$ to 0.01. For evaluation, we employ THR, APS, and RAPS with $\alpha = 0.1$.

Our results in Figure 4 illustrate the superior performance of our approach. For APS and RAPS, both C-Adapter and ConfTr improve the efficiency of conformal predictors, with C-Adapter demonstrating superior performance. Furthermore, C-Adapter enhances the efficiency of THR, whereas ConfTr does not. Additionally, we apply C-Adapter to models that have already been fine-tuned using ConfTr. The results indicate that our approach can further improve the performance of ConfTr. Notably, Baseline+C-Adapter outperforms ConfTr+C-Adapter, suggesting that the accuracy decline associated with ConfTr limits the efficiency of conformal predictors. Overall, empirical results demonstrate that C-Adapter not only surpasses ConfTr but can also enhance its performance.

**Ablation study on the loss function** The size loss from ConfTr can also be utilized to tune our conformal adapter. We conduct an ablation study on ImageNet using DN121, comparing C-Adapter with size loss against our proposed loss function. For the size loss, we maintain a consistent experimental setup and tune the parameter $T$ within the range $\{0.0001, 0.001, 0.01, 0.1\}$, while setting the error rate $\alpha$ to 0.01 during training. For evaluation, we use THR and APS at various coverage rates.

Our results in Table 2 indicate that C-Adapter effectively integrates with size loss, enhancing the efficiency of conformal predictors regardless of the employed score function. However, our pro-

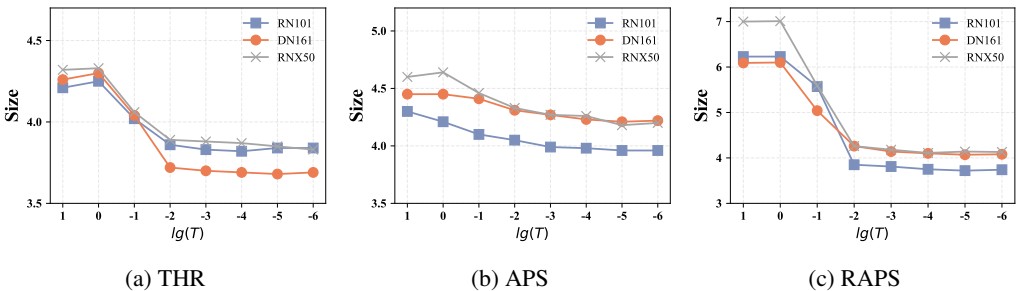

Figure 7: Effect of $T$ on the efficiency of prediction sets with (a) THR, (b) APS, and (c) RAPS.

posed loss function achieves superior average performance. Notably, size loss performs poorly at small error rates $\alpha$; it exhibits inferior performance compared to the baseline when utilizing THR at $\alpha = 0.01$ and $\alpha = 0.02$, while our method consistently outperforms the baseline. Overall, this analysis highlights the flexibility of C-Adapter and the efficacy of our proposed loss function.

**Ablation study on the adaptation strategy** To further demonstrate the significance of this adapter-based tuning method, this ablation compares our approach with two alternative strategies: (1) *Retraining*, which involves training the classifier from scratch with our proposed loss function, and (2) *Fine-tuning*, where the classifier is initially trained with cross-entropy loss and subsequently fine-tuned only on the fully connected layer with our loss. The second strategy is analogous to the fine-tuned version of ConfTr, but it employs a different loss function. In our approach, we first train the classifier using cross-entropy loss and then adapt it for conformal prediction with C-Adapter. This ablation employs a consistent loss function to ensure a fair comparison among different strategies. We provide the detailed setup for the competing methods in Appendix I.2.

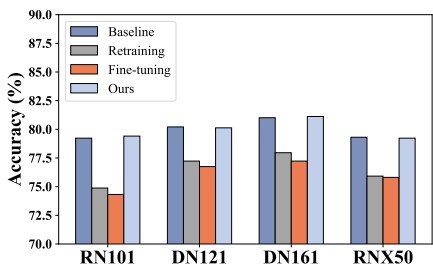

Figure 6: **Accuracy of various adaptation strategies,** on CIFAR-100. Both *Retraining* and *Fine-tuning* result in 3-5% lower accuracy compared to the baseline.

As demonstrated in Figure 6, both *Retraining* and *Fine-tuning* result in 3-5% lower accuracy compared to the baseline. Our results in Figure 5 empirically demonstrate that this decline in accuracy limits overall efficiency: while all three adaptation strategies can enhance the efficiency of APS and RAPS, our method significantly outperforms the others. The negative impact of decreased accuracy is particularly evident in THR, where only our method achieves an improvement in efficiency. Overall, this ablation study further highlights the superiority of our adaptation strategy.

**How does the parameter $T$ affect the performance of C-Adapter?** In Figure 7, we ablate how the parameter $T$ introduced by the surrogate function affects the efficiency of conformal predictors, using THR, APS, and RAPS. This experiment is conducted on ImageNet with RN101, DN161, and RNX50. We set the error rate $\alpha$ to 0.05. As demonstrated in this figure, C-Adapter with a sufficiently small $T$ (below 0.01) stably enhances the efficiency of conformal predictors. This is because the sigmoid function in Equation (9) approximates the indicator function when $T$ is small. For simplicity, we set $T = 10^{-4}$ on ImageNet throughout the experiments.

**C-Adapter shows robustness to distribution shifts.** We investigate the robustness of C-Adapter to distribution shifts. Specifically, we tune C-Adapter using the training set of ImageNet and split ImageNet-V2 into two equal-sized calibration and test sets. Notably, the shifts happen between the training set and calibration/test sets. Thus, coverage will not be affected, as the calibration and test sets remain exchangeable. We examine the performance of C-Adapter on APS, THR, and RAPS at $\alpha = 0.1$ and $\alpha = 0.2$. As demonstrated in Table 3, C-Adapter consistently reduces Size across various base classifiers on ImageNet-V2, regardless of the score function or the predefined error rate $\alpha$. For example, when evaluated on DN161 with $\alpha = 0.1$, C-Adapter reduces the Size of APS from

Table 3: **The robustness of C-Adapter to distribution shift.** C-Adapter is tuned using ImageNet and tested on ImageNet-V2. Since each entry achieves the desired coverage, only **Size** is presented.

| Model | w/o C-Adapter \ w/ C-Adapter | | | | | |
| --- | --- | --- | --- | --- | --- | --- |
| | THR | | APS | | RAPS | |
| | $\alpha$ =0.1 | $\alpha$ =0.2 | $\alpha$ =0.1 | $\alpha$ =0.2 | $\alpha$ =0.1 | $\alpha$ =0.2 |
| RN101 | 6.03 \ **5.43** | 2.11 \ **2.01** | 19.65 \ **5.59** | 7.17 \ **2.57** | 10.90 \ **7.01** | 5.67 \ **2.29** |
| DN121 | 8.01 \ **7.70** | 2.60 \ **2.52** | 24.73 \ **8.14** | 9.13 \ **3.21** | 14.31 \ **10.38** | 7.20 \ **3.01** |
| DN161 | 5.41 \ **4.72** | 2.06 \ **1.91** | 19.32 \ **5.21** | 6.31 \ **2.52** | 10.27 \ **5.98** | 5.18 \ **2.18** |
| RNX50 | 6.80 \ **5.78** | 2.07 \ **2.05** | 26.27 \ **6.11** | 8.58 \ **2.63** | 11.43 \ **7.83** | 6.14 \ **2.38** |
| CLIP | 5.66 \ **5.59** | 2.31 \ **2.29** | 20.73 \ **14.88** | 8.21 \ **6.60** | 10.60 \ **8.67** | 6.35 \ **6.10** |
| Average | 6.38 \ **5.84** | 2.23 \ **2.16** | 22.14 \ **7.99** | 7.88 \ **3.51** | 11.50 \ **7.97** | 6.11 \ **3.19** |

19.32 to 5.21. The results highlight the robustness of C-Adapter to distribution shifts. We further investigate the robustness of C-Adapter under different kinds of data shifts in Appendix K.

**Can C-Adapter be implemented using a hold-out set?** In the original setup, we train C-Adapter using the training set of the classifier $f$. In this study, we investigate whether C-Adapter can be implemented using a hold-out set. Specifically, we randomly divide the CIFAR-100 test set into three subsets: 5,000 samples for training, 3,000 samples for calibration, and 2,000 samples for testing. C-Adapter is tuned for 15 iterations using the 5,000-sample training set with Adam, a batch size of 256, and a learning rate of 0.1, while the parameter $T$ is set to 0.0001. For evaluation, we use THR across different coverage rates ranging from 0.90 to 0.99. We also provide the result for APS in Appendix K.

Our results in Figure 8 demonstrate that C-Adapter consistently achieves improved performance when trained using a hold-out set, regardless of the spec-

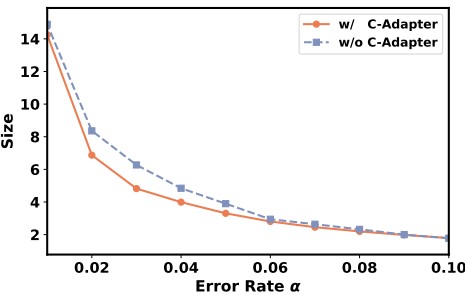

Figure 8: Performance of C-Adapter implemented using a hold-out set on CIFAR-100 with RN101. The efficiency of THR is reported w/ and w/o C-Adapter for comparison.

ified coverage rate. This experiment highlights the adaptability of C-Adapter, which can be effectively implemented with either training data or hold-out data. For scenarios prioritizing data efficiency, C-Adapter can be implemented using the original training data. However, in cases where the pre-trained model exhibits severe overfitting to the training set—a common issue with small-scale datasets—leveraging hold-out data may become essential to enhance generalization.

## 5 CONCLUSION

In this paper, we introduce C-Adapter, an adapter-based tuning method to enhance the efficiency of conformal predictors. Our key idea is to adapt the trained classifiers for conformal prediction while preserving the ranking of labels in the output logits, thereby maintaining the top-$k$ accuracy of the classifiers. To achieve this, we implement the adapter as a class of intra order-preserving functions. For the optimization of C-Adapter, we propose a loss function that enhances the discriminability of non-conformity scores between correctly and randomly matched data-label pairs. Extensive experiments demonstrate that C-Adapter effectively adapts various classifiers for efficient prediction sets and enhances the conformal training method. Our method is user-friendly, as it does not require heavy tuning of hyperparameters and computationally efficient. We hope the insights from this work will inspire future research to explore more effective adaptation strategies for conformal prediction.

**Limitation** Although our adaptation strategy demonstrates promise, we focus solely on using it to optimize the efficiency of conformal predictors. Developing targeted loss functions to adapt deep classifiers for other aspects of conformal prediction (e.g., conditional coverage or robustness) is not explored in this work and offers an interesting direction for future research.

ACKNOWLEDGMENTS

This research is supported by the Shenzhen Fundamental Research Program (Grant No. JCYJ20230807091809020) and the SUSTech-NUS Joint Research Program. Huiping Zhuang is supported by the National Natural Science Foundation of China (Grant No. 62306117) and the Guangzhou Basic and Applied Basic Research Foundation (Grant No. 2024A04J3681). Chi-Man Vong is supported in part by the Shenzhen Science and Technology Innovation Committee (External project no. SGDX20220530111001006) and the University of Macau under MYRG-GRG2023-00061-FST-UMDF. We gratefully acknowledge the support of the Center for Computational Science and Engineering at the Southern University of Science and Technology for our research.

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

## A  RELATED WORK

Conformal prediction has found diverse applications across various domains, including classification (Sadinle et al., 2019), regression (Romano et al., 2019), and more specialized areas such as large language models (Su et al., 2024; Cherian et al., 2024), graph neural networks (Zargarbashi et al., 2023), image generative models (Horwitz & Hoshen, 2022), hyperspectral imaging (Liu et al., 2024), robotic control (Wang et al., 2023), and autonomous systems (Lindemann et al., 2024). In this work, we focus on the split conformal prediction framework (Vovk et al., 2005; Angelopoulos & Bates, 2021), where the training and calibration sets are disjoint. Despite significant progress in developing score functions, such as THR (Sadinle et al., 2019), APS (Romano et al., 2020), RAPS (Angelopoulos et al., 2020), SAPS (Huang et al., 2024a), and RANK (Luo & Zhou, 2024), conformal prediction is typically applied as a post-hoc process for trained classifiers. This separate processing can lead to suboptimal efficiency of conformal predictors.

**Adapting deep classifiers for conformal prediction**  To address the aforementioned issue, several works propose training (fine-tuning) time regularizations to improve the performance of conformal predictors (Stutz et al., 2022; Einbinder et al., 2022; Correia et al., 2024; Huang et al., 2024b). The uncertainty-aware conformal loss function (Einbinder et al., 2022) optimizes the performance of conformal predictors by encouraging the non-conformity scores to follow a uniform distribution, specifically focusing on optimizing APS. To optimize the classifier for maximum predictive efficiency, ConfTr (Stutz et al., 2022) modifies the training objective by introducing a regularization term that minimizes the average set size at a specific error rate. However, this term can negatively impact accuracy by making it challenging to converge to an optimal solution, thereby limiting the overall efficiency of the conformal predictor. Similar works (Huang et al., 2024b; Correia et al., 2024) adopt the ConfTr framework to enhance the efficiency of conformal predictors, yet they still encounter the limitations of ConfTr. Motivated by this, we propose C-Adapter, which enables the efficient adaptation of trained classifiers for conformal prediction without sacrificing accuracy.

**Adapters in other tasks**  Adapters have been extensively explored in parameter-efficient fine-tuning (Houlsby et al., 2019; Rebuffi et al., 2017), aiming to reduce the storage and computational costs of adapting pre-trained models to downstream tasks. These typically involve small, trainable layers integrated into the existing pre-trained model while keeping the original parameters frozen. For instance, LoRA (Hu et al., 2021) has become a standard approach for adapting large language models. Adapters have demonstrated effectiveness across diverse domains (Stickland & Murray, 2019; Sung et al., 2022; Zhang et al., 2023). While C-Adapter aligns with the general concept of adapters, its core insight is fundamentally different. Specifically, C-Adapter introduces an adapter layer to the output layer of the original pre-trained classifier, unlike previous adapters that rely on trainable mid-layers within the model. A distinctive feature of C-Adapter is its ability to preserve the original label ranking, a design uniquely tailored for conformal prediction. The adapters in other tasks cannot preserve the label ranking, making it suboptimal for conformal prediction. This critical difference sets C-Adapter apart from existing adapter techniques in other tasks.

## B  APPLICATION OF C-ADAPTER

The application of C-Adapter is illustrated in Figure 9. C-Adapter refines the raw logits of trained classifiers for conformal prediction while preserving their intra-order, resulting in more efficient conformal prediction sets without compromising the marginal coverage rate.

## C  VITAL TECHNIQUES IN CONFORMAL PREDICTION

### C.1  KEY SCORE FUNCTIONS

Score functions play a crucial role in conformal prediction. With a fixed underlying classifier, the usefulness of the prediction sets is entirely dependent on the chosen score function. Thresholding (THR) (Sadinle et al., 2019) is a commonly used one, which is formulated as:

$$S_{\text{THR}}(\boldsymbol{x}, y; \hat{\pi}) = 1 - \hat{\pi}_y(\boldsymbol{x}).$$

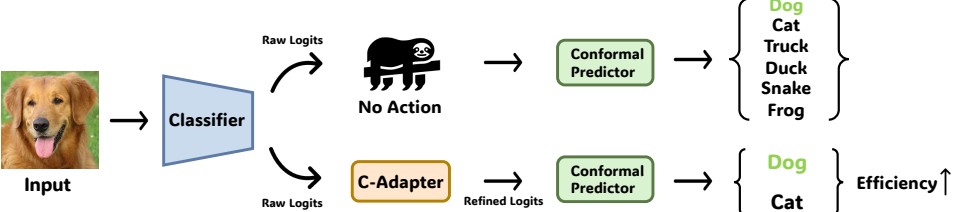

Figure 9: **Application of C-Adapter.** C-Adapter adapts trained classifiers for conformal prediction while preserving the ranking of labels in the output logits. Compared to using the raw logits, this refinement improves the efficiency of prediction sets while maintaining the marginal coverage rate.

THR tends to generate efficient prediction sets. However, this score function frequently undercovers hard examples while overcovering trivial ones, resulting in high conditional coverage violations.

To mitigate this issue, a popular alternative is the series of adaptive prediction sets. Adaptive Prediction Sets (APS) (Romano et al., 2020), the pioneering work in this series, was specifically designed to reduce conditional coverage violations in classification tasks. It is formulated as follows:

$$S_{\text{APS}}(\boldsymbol{x}, y, u; \hat{\pi}) = \sum_{y_i \in \mathcal{Y}} \hat{\pi}_{y_i}(\boldsymbol{x}) \cdot \mathbb{1}_{\{\hat{\pi}_{y_i}(\boldsymbol{x}) > \hat{\pi}_y(\boldsymbol{x})\}} + u \cdot \hat{\pi}_y(\boldsymbol{x}),$$

where $u$ is an independent random variable following a uniform distribution on $[0, 1]$. The prediction set is constructed by adding classes in descending order of probabilities, starting from the most likely to the least, until the cumulative probability exceeds $1 - \alpha$.

However, APS always results in large prediction sets since tail classes with low probabilities are easily included. To alleviate this limitation, Regularized Adaptive Prediction Sets (RAPS) (Angelopoulos et al., 2020) penalizes classes based on their rank information with a predefined threshold, thereby promoting the formation of efficient prediction sets. RAPS is formulated as follows:

$$S_{\text{RAPS}}(\boldsymbol{x}, y, u; \hat{\pi}) = S_{\text{APS}}(\boldsymbol{x}, y, u; \hat{\pi}) + \lambda \cdot (o(y, \hat{\pi}(\boldsymbol{x})) - k_{reg})^+,$$

where $o(y, \hat{\pi}(\boldsymbol{x}))$ is the label ranking of $y$, $\lambda$ and $k_{reg}$ are hyperparameters, and $(z)^+$ denotes the positive part of $z$. This regularization encourages more efficient conformal prediction sets. In this work, we evaluate the performance of C-Adapter on THR, APS, and RAPS. For RAPS, we fix its parameters and consistently set $k_{reg}$ to 1 and $\lambda$ to 0.001 across all experiments.

## C.2 CONFORMAL TRAINING

The core concept of ConfTr (Stutz et al., 2022) is to render the entire conformal prediction pipeline differentiable, thereby enabling direct optimization of the average prediction set size during classifier training. This process involves simulating both the calibration and prediction phases in each mini-batch. Specifically, mini-batch $\mathcal{B}$ is divided into a calibration subset $\mathcal{B}_{\text{cal}}$ and a test subset $\mathcal{B}_{\text{test}}$. The subset $\mathcal{B}_{\text{cal}}$ is used to compute the soft threshold $\tau^{\text{soft}}$, while $\mathcal{B}_{\text{test}}$ is used to obtain the soft prediction sets $\mathcal{C}_{\text{soft}}(\boldsymbol{x}; \tau^{\text{soft}}, \hat{\pi})$ for loss calculations. The detailed operations are as follows:

**Soft threshold:** During the calibration step, a non-differentiable quantile operation is required to determine the threshold $\tau$. To make this operation differentiable, smooth sorting techniques (Blondel et al., 2020; Cuturi et al., 2019; Petersen et al., 2021) are employed, as follows:

$$\tau_\alpha^{\text{soft}} = \mathcal{Q}_{\text{soft}}(\{S(\boldsymbol{x}, y; \hat{\pi})\}_{(\boldsymbol{x}, y) \in \mathcal{B}_{\text{cal}}}, 1 - \alpha), \tag{11}$$

where $\mathcal{Q}_{\text{soft}}$ denotes the differentiable quantile operator, derived using smooth sorting techniques.

**Soft conformal prediction set:** The calculation of conformal prediction sets involves a non-differentiable hard-thresholding operation, as shown in Equation (2). To address this limitation, ConfTr employs the sigmoid function as a differentiable surrogate for the thresholding:

$$\mathcal{C}_{\text{soft}}(\boldsymbol{x}; \tau_\alpha^{\text{soft}}, \hat{\pi}) = \left\{ \sigma\left( \frac{\tau_\alpha^{\text{soft}} - S(\boldsymbol{x}, y; \hat{\pi})}{T} \right) | y \in \mathcal{Y} \right\}, \tag{12}$$

where $\sigma$ denotes the sigmoid function and $T$ is a hyperparameter. The $k$-th term in this set represents a soft assignment of class $k$, indicating the probability of class $k$ being included in the prediction set. By taking the limit as $T \to 0$, this operator becomes

$$\lim_{T \to 0} \sigma \left( \frac{\tau_\alpha^{\text{soft}} - S(\boldsymbol{x}, y; \hat{\pi})}{T} \right) = \begin{cases} 1, & S(\boldsymbol{x}, y; \hat{\pi}) \leq \tau_\alpha^{\text{soft}}, \\ 0, & S(\boldsymbol{x}, y; \hat{\pi}) > \tau_\alpha^{\text{soft}}. \end{cases}$$

For loss calculation, after $\tau^{\text{soft}}$ is computed using $\mathcal{B}_{\text{cal}}$ as specified in Equation (11), Equation (12) is applied to each instance in $\mathcal{B}_{\text{test}}$ to compute the soft prediction sets. The size of each prediction set is approximated by summing the values in the set $\mathcal{C}_{\text{soft}}(\boldsymbol{x})$, which is optimized during training. Additionally, a standard classification loss, such as cross-entropy loss, is incorporated to enhance classification accuracy. The total loss function is then formulated as follows:

$$\mathcal{L}_{\text{ConfTr}}(f(\boldsymbol{x}; \boldsymbol{\theta}), y, \tau_\alpha^{\text{soft}}) = \mathcal{L}_{\text{cls}}(f(\boldsymbol{x}; \boldsymbol{\theta}), y) + \lambda \mathcal{L}_{\text{size}}(f(\boldsymbol{x}; \boldsymbol{\theta}), \tau_\alpha^{\text{soft}}),$$

where $\mathcal{L}_{\text{cls}}$ represents the classification loss, and $\mathcal{L}_{\text{size}}$ refers to the size loss, which approximates the size of the prediction set at a specific error rate (e.g., 0.01). Here, $\lambda$ controls the strength of $\mathcal{L}_{\text{size}}$.

## D    THEORETICAL ANALYSIS OF CLASSIFICATION ACCURACY ON CONFORMAL PREDICTOR EFFICIENCY

In Figures 1 and 11, we have empirically shown that ConfTr may reduce the top-$k$ accuracy of classifiers. In this section, we formally analyze the impact of top-$k$ accuracy on the lower bound of conformal predictors' efficiency. For notation shorthand, given the classifier $\hat{\pi}$, we define $o(y) \equiv o(y, \hat{\pi}(\boldsymbol{x}; \boldsymbol{\theta}))$ to denote the index of label $y$ in the sorted softmax probabilities for $\boldsymbol{x} \sim \mathcal{P}_{\mathcal{X}}$.

**Proposition 2** (Lower bound). *Let $\hat{\pi}$ be a classifier with top-$J$ classification accuracy $acc_J$, and let the error rate be $\alpha$. Then, the expected size of the conformal prediction set is bounded below by:*

$$\mathbb{E}\left[|\mathcal{C}(X)|\right] \geq \begin{cases} (J+1)(1-\alpha) - J \cdot acc_J, & \text{if } acc_J \leq 1 - \alpha, \\ 1 - \alpha, & \text{if } acc_J > 1 - \alpha. \end{cases}$$

*Proof.* To obtain the lower bound of the expected set size, we assume an oracle score function and an ideal model such that, for the case $acc_J \leq 1 - \alpha$:

$$|\mathcal{C}^*(X)| = \begin{cases} 0, & \text{if } Y \notin \mathcal{C}(X), \\ 1, & \text{if } o(Y) \leq J \text{ and } Y \in \mathcal{C}(X), \\ J+1, & \text{if } o(Y) > J \text{ and } Y \in \mathcal{C}(X), \end{cases}$$

where $(X, Y) \sim \mathcal{P}_{\mathcal{X}\mathcal{Y}}$.

Given the top-J accuracy $acc_J$, to satisfy the desired coverage rate of $1 - \alpha$, the expected value of minimal set size is:

$$\mathbb{E}[C^*(X)] = acc_J \cdot 1 + (1 - \alpha - acc_J)(J + 1) + (1 - \alpha) \cdot 0$$
$$= (J + 1)(1 - \alpha) - J \cdot acc_J.$$

Thus, we have the lower bound in the case $acc_J \leq 1 - \alpha$:

$$\mathbb{E}[C(X)] \geq (J + 1)(1 - \alpha) - J \cdot acc_J.$$

Similarly, for the case $acc_J > 1 - \alpha$, we have:

$$|\mathcal{C}^*(X)| = \begin{cases} 0, & \text{if } Y \notin \mathcal{C}(X), \\ 1, & \text{if } o(Y) \leq J \text{ and } Y \in \mathcal{C}(X), \end{cases}$$

where $(X, Y) \sim \mathcal{P}_{\mathcal{X}\mathcal{Y}}$.

In this case, the minimal size of expected prediction sets under the top-J accuracy $acc_J$ is:

$$\mathbb{E}[C^*(X)] = (1 - \alpha)(1) + \alpha \cdot 0 = 1 - \alpha.$$

Therefore, we have the lower bound in the case $acc_J > 1 - \alpha$:

$$\mathbb{E}[C(X)] \geq 1 - \alpha.$$

Now we have the lower bound of the expected set size:

$$\mathbb{E}\left[|\mathcal{C}(X)|\right] \geq \begin{cases} (J+1)(1-\alpha) - J \cdot \text{acc}_J, & \text{if } \text{acc}_J \leq 1 - \alpha, \\ 1 - \alpha, & \text{if } \text{acc}_J > 1 - \alpha. \end{cases}$$

$\square$

According to Proposition 2, the lower bound of the expected set size is negatively related to the top-$k$ accuracy. Therefore, the cost of accuracy introduced by ConfTr will increase the lower bound of the expected size, leading to suboptimal performance in efficiency. This highlights the importance of preserving top-k accuracy in the efficiency optimization for conformal prediction.

# E   INTRA ORDER-PRESERVING FUNCTIONS

**Definition 1.** *A function $h : \mathbb{R}^K \to \mathbb{R}^K$ is considered intra order-preserving if, for any vector $\boldsymbol{x} \in \mathbb{R}^K$, the relative ordering of the elements in $\boldsymbol{x}$ is preserved in $h(\boldsymbol{x})$. Formally, $h_i(\boldsymbol{x}) > h_j(\boldsymbol{x})$ (or $h_i(\boldsymbol{x}) = h_j(\boldsymbol{x})$) holds if and only if $\boldsymbol{x}_i > \boldsymbol{x}_j$ (or $\boldsymbol{x}_i = \boldsymbol{x}_j$).*

An intra order-preserving function maintains all ties and inequalities among the input elements. A typical example is the softmax operator presented in Equation (1). The following theorem outlines the necessary and sufficient conditions for constructing continuous intra order-preserving functions.

**Theorem 1** (Rahimi et al. (2020)). *Let $\mathbb{U}^K \subset \{0,1\}^{K \times K}$ denote the set of $K \times K$ permutation matrices, and let $R : \mathbb{R}^K \to \mathbb{U}^K$ represent the sorting function. For any vector $\boldsymbol{x} \in \mathbb{R}^K$, the vector $\boldsymbol{r} = R(\boldsymbol{x})\boldsymbol{x}$ satisfies $\boldsymbol{r}_1 \geq \cdots \geq \boldsymbol{r}_K$. A continuous function $h : \mathbb{R}^K \to \mathbb{R}^K$ is intra order-preserving if and only if it can be written as $h(\boldsymbol{x}) = R(\boldsymbol{x})^{-1} U t(\boldsymbol{x})$, where $U$ is an upper-triangular matrix of ones, and $t : \mathbb{R}^K \to \mathbb{R}^K$ is a continuous function that satisfies the following condition: $t_i(\boldsymbol{x}) > 0$ (or $t_i(\boldsymbol{x}) = 0$) if $\boldsymbol{r}_i > \boldsymbol{r}_{i+1}$ (or $\boldsymbol{r}_i = \boldsymbol{r}_{i+1}$), for all $i < K$. The value of $t_K(\boldsymbol{x})$ is arbitrary.*

This theorem provides a pathway for learning within this function family using backpropagation. To better demonstrate how the transformation in Equation 4 refines input logits for conformal prediction without compromising their ranking, we analyze the algorithm step by step as follows:

1. For an input logit vector $\boldsymbol{f}$, we sort it in descending order. The resulting sorted vector is denoted as $\boldsymbol{r}$, where $\boldsymbol{r}_1 > \boldsymbol{r}_2 > \cdots > \boldsymbol{r}_K$.

2. We calculate $\Psi(\boldsymbol{f})$ using Equation (5). It can be observed that, except for $\Psi_K(\boldsymbol{f})$, if $\boldsymbol{r}_i > \boldsymbol{r}_{i+1}$, this will always result in $\Psi_i(\boldsymbol{f}) > 0$; and if $\boldsymbol{r}_i = \boldsymbol{r}_{i+1}$, it will always result in $\Psi_i(\boldsymbol{f}) = 0$. Essentially, $\Psi(\boldsymbol{f})$ captures the absolute difference between each element in the transformed logits and the next smaller element. **This term is designed to ensure that all ties are preserved in the transformed logits.**

3. After calculating $\Psi(\boldsymbol{f})$, we obtain a sorted vector $U\Psi(\boldsymbol{f})$ by performing a reverse cumulative sum operation, where $U$ is an upper-triangular matrix of ones. This sorted vector is denoted as $\boldsymbol{v} = U\Psi(\boldsymbol{f})$. **We observe that $\boldsymbol{v}_i > \boldsymbol{v}_{i+1}$ (or $\boldsymbol{v}_i = \boldsymbol{v}_{i+1}$) holds if and only if $\boldsymbol{r}_i > \boldsymbol{r}_{i+1}$ (or $\boldsymbol{r}_i = \boldsymbol{r}_{i+1}$).** Notably, $\boldsymbol{v}$ is the sorted version of the refined logits.

4. The reverse sorting operator $R(\boldsymbol{f})^{-1}$ is applied to rearrange $\boldsymbol{v}$, aligning it with the order of $\boldsymbol{f}$. This ensures that the resulting vector preserves all ties and inequalities among the input elements. The expressivity of this transformation is guaranteed by the adaptive layer $\varphi$.

**Additional implementation details with optimization strategies of C-Adapter**

- For the adaptive layer $\varphi(\boldsymbol{f})$, we use a single fully connected layer (linear layer) without structural tuning in our current work.

- The input $\boldsymbol{f}$ is rescaled to the range $(0, 1)$ using the softmax before passing through the adapter, which helps with optimization. While using raw logits or applying log-softmax to transform $\boldsymbol{f}$ can also work, they empirically lead to suboptimal performance.

- A residual term is always incorporated, i.e., $g(\boldsymbol{f}; \boldsymbol{w}) = R(\boldsymbol{f})^{-1} U\Psi(\boldsymbol{f}) + \boldsymbol{f}$, which consistently leads to a better local optimum solution in our experiments.

- Empirically, we observe that using $(r_i - r_{i+1})$ in Equation 5 results in suboptimal performance. In contrast, applying $\sqrt{r_i - r_{i+1}}$ leads to better local optimum and enhanced efficiency. Further gains are achieved with $\{\sqrt[a]{r_i - r_{i+1}}\}$, particularly as $a$ increases, showing significant improvements on CIFAR-100. A stable empirical solution is to set $a$ to $\infty$, which corresponds to directly applying $\mathbb{1}_{\{r_i > r_{i+1}\}}$ in Equation 5.

## F   PROOF FOR PROPOSITION 1

*Proof.* Considering $(X,Y) \sim \mathcal{P}_{\mathcal{X}\mathcal{Y}}, \hat{X} \sim \mathcal{P}_{\mathcal{X}}, \hat{Y} \sim \text{Uniform}(\mathcal{Y})$, and letting $\mu(\hat{\pi}) := \mathbb{P}\left(S(X,Y;\hat{\pi}) \geq S(\hat{X},\hat{Y};\hat{\pi})\right)$, we have

$$\mu(\hat{\pi}) = \mathbb{E}_{(X,Y)\sim\mathcal{P}_{\mathcal{X}\mathcal{Y}}, \hat{X}\sim\mathcal{P}_{\mathcal{X}}, \hat{Y}\sim\text{Uniform}(\mathcal{Y})}\left[\mathbb{1}_{\{S(X,Y;\hat{\pi})\geq S(\hat{X},\hat{Y};\hat{\pi})\}}\right]$$

$$= \mathbb{E}_{\hat{X}\sim\mathcal{P}_{\mathcal{X}}, \hat{Y}\sim\text{Uniform}(\mathcal{Y})}\left[\mathbb{E}_{(X,Y)\sim\mathcal{P}_{\mathcal{X}\mathcal{Y}}}\left[\mathbb{1}_{\{S(X,Y;\hat{\pi})\geq S(\hat{X},\hat{Y};\hat{\pi})\}}\right]\right]$$

$$= \frac{1}{K}\sum_{\hat{Y}\in\mathcal{Y}}\mathbb{E}_{\hat{X}\sim\mathcal{P}_{\mathcal{X}}}\left[\mathbb{E}_{(X,Y)\sim\mathcal{P}_{\mathcal{X}\mathcal{Y}}}\left[\mathbb{1}_{\{S(X,Y;\hat{\pi})\geq S(\hat{X},\hat{Y};\hat{\pi})\}}\right]\right]$$

$$= \frac{1}{K}\mathbb{E}_{\hat{X}\sim\mathcal{P}_{\mathcal{X}}}\left[\mathbb{E}_{(X,Y)\sim\mathcal{P}_{\mathcal{X}\mathcal{Y}}}\left[\sum_{\hat{Y}\in\mathcal{Y}}\mathbb{1}_{\{S(X,Y;\hat{\pi})\geq S(\hat{X},\hat{Y};\hat{\pi})\}}\right]\right]$$

$$= \frac{1}{K}\mathbb{E}_{\hat{X}\sim\mathcal{P}_{\mathcal{X}}}\left[\mathbb{E}_{s_\theta\sim\mathcal{P}_{S_\theta}}\left[\sum_{\hat{Y}\in\mathcal{Y}}\mathbb{1}_{\{s_\theta\geq S(\hat{X},\hat{Y};\hat{\pi})\}}\right]\right].$$

Assuming that the CDF of $\mathcal{P}_\theta$, denoted as $F_{S_\theta}$, is monotonically increasing, we have

$$\mathbb{E}_{\hat{X}\sim\mathcal{P}_{\mathcal{X}}}\left[\mathbb{E}_{s_\theta\sim\mathcal{P}_{S_\theta}}\left[\sum_{\hat{Y}\in\mathcal{Y}}\mathbb{1}_{\{s_\theta\geq S(\hat{X},\hat{Y};\hat{\pi})\}}\right]\right]$$

$$= \mathbb{E}_{\hat{X}\sim\mathcal{P}_{\mathcal{X}}}\left[\int\sum_{\hat{Y}\in\mathcal{Y}}\mathbb{1}_{\{t\geq S(\hat{X},\hat{Y};\hat{\pi})\}}\mathrm{d}F_{S_\theta}(t)\right]$$

$$(\text{let } t = F_{S_\theta}^{-1}(1-\alpha)) = \mathbb{E}_{\hat{X}\sim\mathcal{P}_{\mathcal{X}}}\left[\int_1^0\sum_{\hat{Y}\in\mathcal{Y}}\mathbb{1}_{\{F_{S_\theta}^{-1}(1-\alpha)\geq S(\hat{X},\hat{Y};\hat{\pi})\}}\mathrm{d}(1-\alpha)\right]$$

$$= \mathbb{E}_{\hat{X}\sim\mathcal{P}_{\mathcal{X}}}\left[\int_0^1\sum_{\hat{Y}\in\mathcal{Y}}\mathbb{1}_{\{F_{S_\theta}^{-1}(1-\alpha)\geq S(\hat{X},\hat{Y};\hat{\pi})\}}\mathrm{d}\alpha\right]$$

$$= \mathbb{E}_{\hat{X}\sim\mathcal{P}_{\mathcal{X}}}\left[\int_0^1|\mathcal{C}\left(\hat{X}; F_{S_\theta}^{-1}(1-\alpha), \hat{\pi}\right)|\,\mathrm{d}\alpha\right].$$

Thus, $\mu(\hat{\pi}) > \mu(\hat{\pi}')$ if and only if

$$\mathbb{E}_{X\sim\mathcal{P}_{\mathcal{X}}}\left[\int_0^1|\mathcal{C}\left(X; F_{S_\theta}^{-1}(1-\alpha), \hat{\pi}\right)|\,\mathrm{d}\alpha\right] > \mathbb{E}_{X\sim\mathcal{P}_{\mathcal{X}}}\left[\int_0^1|\mathcal{C}\left(X; F_{S_\theta}^{-1}(1-\alpha), \hat{\pi}'\right)|\,\mathrm{d}\alpha\right].$$

$\square$

## G   CONVERGENCE ANALYSIS

Our method is computationally efficient, as it updates only a limited number of parameters and converges rapidly. To demonstrate this, we conduct an experiment on ImageNet using DN121, visualizing the changes in average loss and efficiency over iterations. The C-Adapter is tuned with

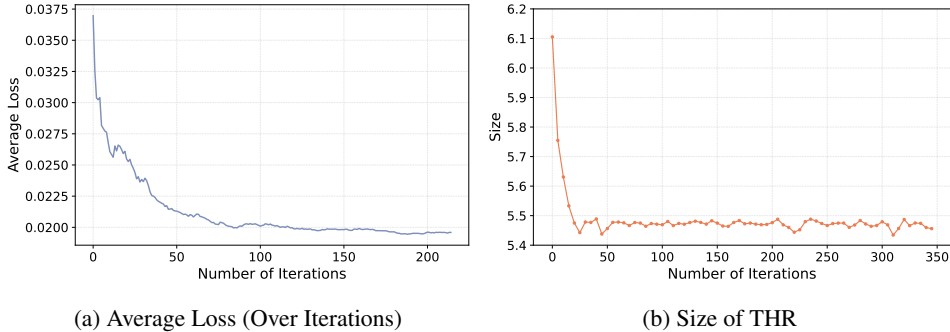

(a) Average Loss (Over Iterations)            (b) Size of THR

Figure 10: Convergence analysis of C-Adapter on ImageNet using DN121.

Adam, a learning rate of 0.1, a batch size of 256, and a weight decay of 0.0001. THR is applied during both adapter tuning and evaluation, with the error rate set to 0.5. As shown in Figure 10, our method converges rapidly within 200 iterations, with the efficiency of the conformal predictor improving quickly and approaching nearly optimal performance within just 50 iterations. (Note that the average loss is calculated by averaging the loss across batches over iterations.) This convergence analysis highlights the computational efficiency of our proposed approach.

## H  EVALUATION METRICS

Size refers to the average number of labels in the prediction sets, Size is defined by:

$$\text{Size} = \frac{1}{|\mathcal{D}_{\text{test}}|} \sum_{(\boldsymbol{x},y)\in\mathcal{D}_{\text{test}}} |\mathcal{C}(\boldsymbol{x})|.$$

Coverage refers to the percentage of test samples for which the prediction sets include the ground-truth labels, and is defined as:

$$\text{Coverage} = \frac{1}{|\mathcal{D}_{\text{test}}|} \sum_{(\boldsymbol{x},y)\in\mathcal{D}_{\text{test}}} \mathbb{1}_{\{y\in\mathcal{C}(\boldsymbol{x})\}}.$$

## I  DETAILED EXPERIMENTAL SETUP

### I.1  DETAILED SETUP FOR FIGURE 1

For CIFAR100, ResNet18 is trained using the training set of 50,000 samples. The test set of 10,000 samples is divided into a calibration subset of 5,000 samples and a test subset of 5,000 samples. The calibration subset is further split into a validation set and a calibration set in an 20:80 ratio for parameter tuning. The network is trained for 200 epochs using SGD with a momentum of 0.9, a weight decay of 0.0005, and a batch size of 256. The initial learning rate is set to 0.1 and is reduced by a factor of 5 at 60, 120, and 160 epochs. The hyperparameters $T$ and $\lambda$ of ConfTr are tuned from the ranges $\{0.01, 0.1, 0.5, 1\}$ and $\{0.005, 0.01, 0.05, 0.1, 0.2\}$, respectively.

For ImageNet, instead of training the classifier from scratch, we fine-tune only the fully connected layer of a pre-trained ResNet18 using the training set. This is also a commonly applied setting of ConfTr (Stutz et al., 2022). The test set is divided into a calibration subset of 30,000 samples and a test subset of 20,000 samples, with the calibration subset further split into a validation set and a calibration set in a 20:80 ratio for parameter tuning. The fully connected layer is tuned for 240 iterations using Adam with a batch size of 256 and a learning rate of 0.001. A larger learning rate significantly decreases classification accuracy, thereby reducing efficiency. The hyperparameters $T$ and $\lambda$ for ConfTr are selected from the ranges $\{0.01, 0.1, 0.5, 1\}$ and $\{0.001, 0.005, 0.01, 0.05, 0.1\}$, respectively; a larger $\lambda$ also leads to a substantial decline in accuracy in our experiments.

For evaluation, we use THR, APS, and RAPS, with the error rate $\alpha$ set to 0.1. For RAPS, we fix its parameters and consistently set $k_{reg}$ to 1 and $\lambda$ to 0.001. During model training, we utilize the THRLP score function (Stutz et al., 2022), setting the error rate $\alpha$ to 0.01. We also present the top-2, top-3, and top-5 accuracy of ConfTr on CIFAR100 and ImageNet in Figure 11.

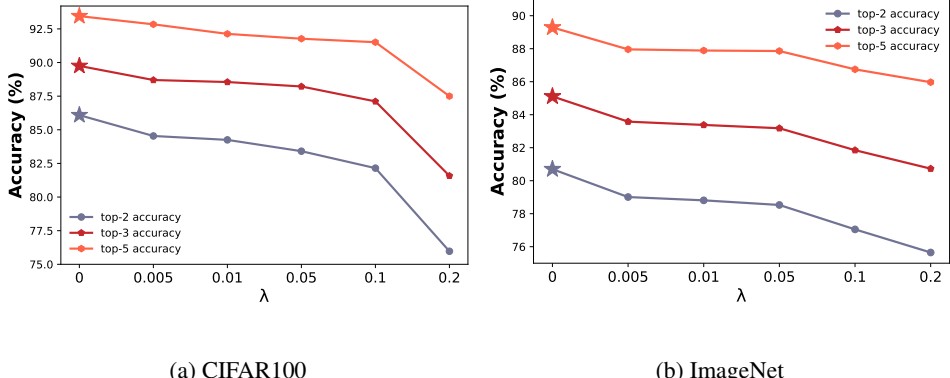

(a) CIFAR100             (b) ImageNet

Figure 11: **The accuracy of ConfTr with various λ**, using ResNet18 on (a) CIFAR-100 and (b) ImageNet. ★ denotes the baseline without ConfTr. The results indicate that increasing λ consistently decreases the top-2, top-3, and top-5 classification accuracies.

### I.2    DETAILED SETUP FOR FIGURE 5

***Retraining***: Classifiers are trained using the complete training set of 50,000 samples, with the objective defined in Equation (9). The network is trained for 200 epochs using SGD with a momentum of 0.9, a weight decay of 0.0005, and a batch size of 256. The initial learning rate is set to 0.1 and reduced by a factor of 5 at epochs 60, 120, and 160. The parameter $T$ is tuned within the range $\{0.001, 0.01, 0.1, 1\}$ using the validation set. We utilize THR for classifier training.

***Fine-tuning***: Classifiers are trained using the same training set of 50,000 samples with cross-entropy loss. The network is trained for 200 epochs using SGD, with a momentum of 0.9, a weight decay of 0.0005, and a batch size of 256. The initial learning rate is set to 0.1, reduced by a factor of 5 at epochs 60, 120, and 160. During fine-tuning, only the fully connected layer of the pre-trained classifier is updated, training for 240 iterations with Adam, a batch size of 256, and a learning rate of 0.001. Notably, a larger learning rate results in a significant decrease in classification accuracy. The parameter $T$ is tuned from the range $\{0.001, 0.01, 0.1, 1\}$ using the validation set.

## J    ANALYSIS OF CONDITIONAL COVERAGE

**Evaluation metrics**    We examine the impact of C-Adapter on conditional coverage and consider two metrics: (1) class-conditional coverage gap (CovGap) (Ding et al., 2024) and (2) size-stratified coverage violation (SSCV) (Angelopoulos et al., 2020) on ImageNet. CovGap (Ding et al., 2024) and SSCV (Angelopoulos et al., 2020; Huang et al., 2024a) are defined as follows:

$$\text{CovGap} = 100 \times \frac{1}{|\mathcal{Y}|} \sum_{y \in \mathcal{Y}} |\hat{c}_y - (1 - \alpha)|,$$

$$\text{SSCV} = 100 \times \sup_{j} \left| (1 - \alpha) - \frac{|\{i : y_i \in \mathcal{C}\left(\boldsymbol{x}_i\right), i \in \mathcal{J}_j\}|}{|\mathcal{J}_j|} \right|.$$

For CovGap, $\hat{c}_y$ denotes the coverage rate for class $y$ and quantifies the deviation of class-conditional coverage from the desired level of $1 - \alpha$. For SSCV, $\mathcal{J}$ represents the partitioned sets, with prediction sets categorized by their sizes. This metric evaluates the maximum deviation of the observed coverage rate from $1 - \alpha$ across different set size categories. In our experiment, the set size partitioning for SSCV is defined as $\{0\text{-}1, 2, 3, 4, 5, 6, 7, 8, 9, 10, 11\text{-}100, 101\text{-}1000\}$.

Since SSCV evaluates the maximum deviation (as opposed to the mean deviation used by CovGap) of the selected group, extremely small groups may result in disproportionately large coverage gaps, making the metric unreliable. For example, when set $\alpha = 0.1$, if a size group contains only one sample and is not covered, SSCV could yield a value of 90, even though the metric performs well for other groups. To ensure a accurate reflection of the adaptiveness of prediction sets, we apply a threshold to exclude groups with excessively small sizes (less than $1/200$ of the test set size).

Table 4: **Experimental results on conditional coverage.** This experiment is conducted on ImageNet with different architectures. ↓ indicates that lower values are better. Coverage is omitted in this table. C-Adapter consistently reduces Size, SSCV, and CovGap in most cases.

| | | Size ↓ / SSCV ↓ / CovGap ↓ | | | | | |
|---|---|---|---|---|---|---|---|
| | | **RN101** | **DN121** | **DN161** | **RNX50** | **CLIP** | **Average** |
| α = 0.05 | APS | 15.77 / 3.47 / 4.44 | 19.81 / 3.09 / 4.35 | 16.59 / 3.86 / 4.57 | 21.27 / 4.48 / 4.49 | 27.12 / 3.51 / 4.83 | 20.11 / 3.68 / 4.54 |
| | +Ours | **12.99 / 2.91 / 4.34** | **15.55 / 2.26 / 4.33** | **11.25 / 2.66 / 4.50** | **14.11 / 3.57 / 4.44** | **17.64 / 2.61 / 4.81** | **14.31 / 2.80 / 4.48** |
| | RAPS | 7.42 / 3.08 / 4.51 | 10.43/ 3.94 / 4.45 | 7.28 / 3.36 / 4.64 | 7.87 / 3.76 / 4.62 | 15.21 / 1.71 / 5.06 | 9.64 / 3.17 / 4.66 |
| | +Ours | **7.15 / 2.85 / 4.44** | **9.33 / 1.75 / 4.43** | **6.53 / 2.74 / 4.61** | **7.31 / 3.03 / 4.60** | **12.61 / 1.65 / 5.01** | **8.59 / 2.40 / 4.62** |
| α = 0.1 | APS | 6.86 / 6.49 / 6.32 | 9.39 / 5.37 / **6.23** | 6.78 / 6.51 / **6.42** | 8.66 / 7.14 / 6.50 | 13.31 / 6.77 / 7.61 | 9.00 / 6.46 / 6.62 |
| | +Ours | **6.24 / 4.33 / 6.21** | **7.80 / 3.89 /** 6.25 | **5.38 / 4.80 /** 6.52 | **6.72 / 5.12 / 6.41** | **9.34 / 3.73 /** 7.61 | **7.10 / 4.37 / 6.60** |
| | RAPS | 4.77 / 4.69 / 6.37 | 6.62 / 2.52 / 6.33 | 4.59 / 5.16 / **6.53** | 5.15 / **3.77** / 6.60 | 9.21 / 3.39 / 7.50 | 6.07 / 3.90 / 6.67 |
| | +Ours | **4.64 / 4.07 / 6.27** | **5.95 / 3.40 /** 6.34 | **4.17 / 4.55 /** 6.55 | **4.79 /** 3.92 / **6.50** | **7.86 / 2.61 / 7.49** | **5.48 / 3.71 / 6.63** |

Table 5: **Performance of C-Adapter on common benchmarks.** ↓ indicates that a smaller value is better. Results in **bold** indicate superior performance. C-Adapter is tuned using APS.

| | | w/o C-Adapter \ w/ C-Adapter | | | | | | | |
|---|---|---|---|---|---|---|---|---|---|
| | | **ImageNet** | | | | **CIFAR-100** | | | |
| **Score** | **Model** | α = 0.05 | | α = 0.1 | | α = 0.05 | | α = 0.1 | |
| | | Coverage | Size (↓) | Coverage | Size (↓) | Coverage | Size (↓) | Coverage | Size (↓) |
| THR | RN101 | 0.95 \ 0.95 | 4.03 \ **3.66** | 0.90 \ 0.90 | 1.91 \ **1.86** | 0.95 \ 0.95 | 3.64\ **3.11** | 0.90 \ 0.90 | 1.87 \ **1.80** |
| | DN121 | 0.95 \ 0.95 | 5.66 \ **5.39** | 0.90 \ 0.90 | 2.42 \ **2.40** | 0.95 \ 0.95 | 3.27 \ **3.03** | 0.90 \ 0.90 | 1.72 \ **1.69** |
| | DN161 | 0.95 \ 0.95 | 4.03 \ **3.88** | 0.90 \ 0.90 | 1.89 \ **1.86** | 0.95 \ 0.95 | 2.91 \ **2.71** | 0.90 \ 0.90 | 1.72 \ **1.71** |
| | RNX50 | 0.95 \ 0.95 | 4.06 \ **3.93** | 0.90 \ 0.90 | 1.87 \ **1.84** | 0.95 \ 0.95 | 3.41 \ **3.16** | 0.90 \ 0.90 | 1.78 \ **1.77** |
| | CLIP | 0.95 \ 0.95 | **6.88** \ 6.90 | 0.90 \ 0.90 | 3.33 \ **3.28** | 0.95 \ 0.95 | 9.71\ **9.67** | 0.90 \ 0.90 | 4.78 \ **4.69** |
| | **Average** | 0.95 \ 0.95 | 4.93 \ **4.75** | 0.90 \ 0.90 | 2.29 \ **2.25** | 0.95 \ 0.95 | 4.59 \ **4.34** | 0.90 \ 0.90 | 2.37 \ **2.33** |
| APS | RN101 | 0.95 \ 0.95 | 14.73 \ **3.82** | 0.90 \ 0.90 | 7.23 \ **2.07** | 0.95 \ 0.95 | 7.60 \ **3.16** | 0.90 \ 0.90 | 3.95 \ **1.80** |
| | DN121 | 0.95 \ 0.95 | 20.00 \ **5.64** | 0.90 \ 0.90 | 9.21 \ **2.74** | 0.95 \ 0.95 | 10.20 \ **4.12** | 0.90 \ 0.90 | 4.44 \ **2.35** |
| | DN161 | 0.95 \ 0.95 | 16.43 \ **4.13** | 0.90 \ 0.90 | 6.82 \ **2.05** | 0.95 \ 0.95 | 9.90 \ **3.14** | 0.90 \ 0.90 | 5.42 \ **1.87** |
| | RNX50 | 0.95 \ 0.95 | 21.54 \ **4.10** | 0.90 \ 0.90 | 8.92 \ **2.07** | 0.95 \ 0.95 | 9.95 \ **3.19** | 0.90 \ 0.90 | 5.14 \ **1.90** |
| | CLIP | 0.95 \ 0.95 | 26.35 \ **7.42** | 0.90 \ 0.90 | 13.24 \ **3.43** | 0.95 \ 0.95 | 16.13 \ **12.94** | 0.90 \ 0.90 | 10.18 \ **8.10** |
| | **Average** | 0.95 \ 0.95 | 19.81 \ **5.04** | 0.90 \ 0.90 | 9.08 \ **2.47** | 0.95 \ 0.95 | 10.76 \ **5.31** | 0.90 \ 0.90 | 6.01 \ **3.20** |
| RAPS | RN101 | 0.95 \ 0.95 | 7.13 \ **4.43** | 0.90 \ 0.90 | 4.60 \ **2.01** | 0.95 \ 0.95 | 5.16 \ **4.71** | 0.90 \ 0.90 | 3.25 \ **1.81** |
| | DN121 | 0.95 \ 0.95 | 10.28 \ **7.38** | 0.90 \ 0.90 | 6.57 \ **2.66** | 0.95 \ 0.95 | 7.19\ **4.00** | 0.90 \ 0.90 | 4.50 \ **1.83** |
| | DN161 | 0.95 \ 0.95 | 7.31 \ **5.01** | 0.90 \ 0.90 | 4.63 \ **2.00** | 0.95 \ 0.95 | 7.10 \ **3.22** | 0.90 \ 0.90 | 4.59 \ **1.81** |
| | RNX50 | 0.95 \ 0.95 | 7.88 \ **5.05** | 0.90 \ 0.90 | 5.20 \ **2.01** | 0.95 \ 0.95 | 7.20 \ **3.64** | 0.90 \ 0.90 | 4.47 \ **1.79** |
| | CLIP | 0.95 \ 0.95 | 15.14 \ **8.74** | 0.90 \ 0.90 | 9.25 \ **3.41** | 0.95 \ 0.95 | 14.52 \ **13.61** | 0.90 \ 0.90 | 9.41 \ **8.92** |
| | **Average** | 0.95 \ 0.95 | 9.55 \ **6.12** | 0.90 \ 0.90 | 6.05 \ **2.42** | 0.95 \ 0.95 | 8.24 \ **5.84** | 0.90 \ 0.90 | 5.24 \ **3.23** |

**Experiment** In this study, we further demonstrate that C-Adapter can improve or maintain the conditional coverage metrics of both APS and RAPS with enhanced efficiency. For the main experiment in Table 1, we apply early stopping and select the iteration with optimal efficiency at the desired coverage rate. In this experiment, all experimental setups remain consistent, except that efficiency-oriented early stopping is not applied. Instead, training continues until the loss converges (720 iterations) to assess the impact of C-Adapter on conditional coverage metrics.

As detailed in Table 4, C-Adapter consistently reduces Size, SSCV, and CovGap in most cases tested on ImageNet. Notably, the reduction in Size is less substantial compared to the results in Table 1. Thus, users may adopt a training strategy that best aligns with their specific needs for efficiency or conditional coverage metrics. Overall, this experiment shows that C-Adapter can enhance or maintain the conditional coverage of APS and RAPS while simultaneously improving their efficiency, highlighting its flexibility in improving the efficiency of conformal predictors.

# K ADDITIONAL EXPERIMENTAL RESULTS

**Results when tuning C-Adapter using APS** We present the detailed results for Coverage and Size when tuning C-Adapter using APS. Empirical results in Table 5 show that C-Adapter consistently improves the efficiency of conformal predictors, regardless of model architecture or pre-training

Table 6: **Robustness of C-Adapter under different data shifts.** C-Adapter is tuned using ImageNet and tested on ImageNet-R and ImageNet-A. The base classifier adopted in this experiment is RN101. Since each entry achieves the desired coverage, only **Size** is presented.

| | w/o C-Adapter \ w/ C-Adapter | | | | | |
|---|---|---|---|---|---|---|
| | THR | | APS | | RAPS | |
| Dataset | $\alpha = 0.4$ | $\alpha = 0.5$ | $\alpha = 0.4$ | $\alpha = 0.5$ | $\alpha = 0.4$ | $\alpha = 0.5$ |
| ImageNet-R | 6.13 \ **5.56** | 2.23 \ **2.11** | 11.32 \ **7.16** | 6.16 \ **2.61** | 6.24 \ **5.95** | 3.32 \ **2.78** |
| ImageNet-A | 26.99 \ **20.09** | 18.21 \ **13.13** | 36.62 \ **23.01** | 25.98 \ **17.56** | 20.26 \ **20.08** | 13.72 \ **13.40** |
| Average | 16.56 \ **13.25** | 10.22 \ **7.62** | 23.97 \ **15.09** | 16.07 \ **10.09** | 13.25 \ **13.02** | 8.52 \ **8.09** |

Table 7: **Performance of C-Adapter on text classification.** ↓ indicates that a smaller value is better. C-Adapter is tuned using THR. The experiment is carried on dbpedia_14 using LLama3-8B.

| | w/o C-Adapter \ w/ C-Adapter | | | |
|---|---|---|---|---|
| Score | $\alpha = 0.05$ | | $\alpha = 0.1$ | |
| | Coverage | Size ($\downarrow$) | Coverage | Size ($\downarrow$) |
| THR | 0.94 \ 0.95 | 2.80 \ **2.61** | 0.89 \ 0.89 | 2.17 \ **2.04** |
| APS | 0.95 \ 0.94 | 3.14 \ **2.75** | 0.90 \ 0.91 | 2.33 \ **2.08** |
| RAPS | 0.95 \ 0.95 | 3.23 \ **3.11** | 0.90 \ 0.90 | 2.48 \ **2.32** |
| Average | – | 3.06 \ **2.82** | – | 2.33 \ **2.15** |

strategy, underscoring the flexibility of our approach. Since THR is more efficient to compute, we use it by default in the training of C-Adapter.

**C-Adapter is robust to different distribution shifts.** We further investigate the robustness of C-Adapter on ImageNet-R(rendition) (Hendrycks et al., 2021a) and ImageNet-A(adversarial) (Hendrycks et al., 2021b). ImageNet-A and ImageNet-R are extended versions of the ImageNet dataset designed to evaluate model robustness, with ImageNet-A focusing on adversarial examples that are modified to mislead models, and ImageNet-R consisting of images transformed by various artistic styles and visual changes to test models' adaptability to different visual distributions.

Specifically, we tune C-Adapter using the ImageNet training set, then split both ImageNet-R and ImageNet-A into equal-sized calibration and test sets for conformal prediction. In this experiment, we use pre-trained RN101. Given the relatively low performance of the pre-trained RN101 on ImageNet-R and ImageNet-A (for example, it achieves only 39% classification accuracy on ImageNet-R), we set the error rate $\alpha$ to 0.4 and 0.5, respectively. **Notably, coverage remains unaffected by this setting, as the calibration and test sets are still exchangeable.** We evaluate the performance of C-Adapter using the APS, THR, and RAPS. As shown in Table 6, C-Adapter consistently reduces size across various base classifiers on both ImageNet-R and ImageNet-A, regardless of the score function used in conformal prediction or the pre-defined error rate $\alpha$. The experimental results further highlight the robustness of C-Adapter under different kinds of data shift.

**Results on text classification using LLMs** To provide a comprehensive understanding of the broad capabilities of our method, we evaluate its performance on a text classification task. Additionally, we demonstrate the versatility of C-Adapter by performing this task using Large Language Models (LLMs). Before delving into the experiment, we introduce a commonly used framework for applying LLMs to text classification. For each input, we construct a prompt in the following format:

```
\n Company, Educational Institution, Artist, Athlete, Office
   Holder, Mode of Transportation, Building, Natural Place,
   Village, Animal, Plant, Album, Film, or Written Work? \n Input
   : </text> \n Output: </text>
```

In this prompt, each input instance $x$ is paired with a label $y \in \mathcal{Y}$, where $\mathcal{Y}$ represents the set of possible categories (e.g., "Company", "Artist", etc.). The input $x$ and label $y$ are then inserted into the appropriate position in the prompt. After obtaining the model's output, we construct the

input-output pair $z$, tokenize it, and calculate its perplexity as follows:

$$\text{Perplexity}(\boldsymbol{z}) = \exp\left(-\frac{1}{N}\sum_{i=1}^{N}\log p(z_i|z_{<i})\right),$$

where $p(z_i|z_{<i})$ denotes the predicted probability of token $z_i$ given the preceding tokens, and $N$ is the total number of tokens in the input-output pair $\boldsymbol{z}$. Perplexity quantifies the uncertainty of the language model in predicting the next token in the sequence. The label corresponding to the input-output pair $\boldsymbol{z}$ with the lowest perplexity is selected as the predicted label for the input $\boldsymbol{x}$.

We perform conformal prediction in this classification setting, specifically on the dbpedia_14 dataset, a 14-class classification task (Lehmann et al., 2015). For the LLM, we use LLama3-8B (Touvron et al., 2023). In this experiment, we apply conformal prediction based on perplexity. For an input $\boldsymbol{x}$, we compute the perplexity vector for each label $y \in \mathcal{Y}$. Since a lower perplexity indicates a more likely label, **the reciprocal of the normalized perplexity is used as the raw logits to perform conformal prediction.** We use the test set of dbpedia_14 with 7,480 samples to perform the experiment, where 2,000 of them are used for tuning, and the remaining data is equally and randomly divided into calibration and test sets. Notably, we rely on the zero-shot ability of LLama3-8B to perform classification. The classification accuracy on this set is 66%.

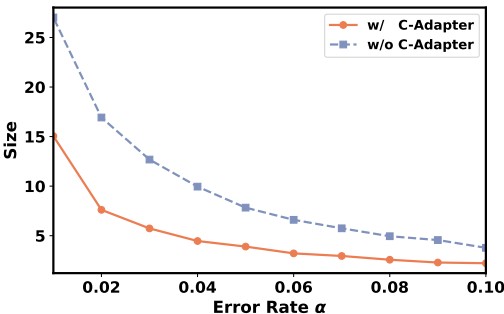

Figure 12: Performance of C-Adapter implemented using a hold-out set on CIFAR-100 with RN101. The efficiency of APS is reported w/ and w/o C-Adapter for comparison.

The experimental results are presented in Table 7. Compared to using the raw logits as input, C-Adapter significantly improves the efficiency of conformal predictor across different conditions.

**Additional results on implementing C-Adapter using hold-out data** Our results in Figure 12 demonstrate that C-Adapter consistently achieves improved performance when trained using a hold-out set, regardless of the specified coverage rate. Unlike the results in Figure 8 of the main paper, this experiment uses the APS non-conformity score function while keeping all other settings unchanged.

