# OpenReview forum: "C-Adapter: Adapting Deep Classifiers for Efficient Conformal Prediction Sets"
_ICLR.cc/2025/Conference — ICLR 2025 Conference Withdrawn Submission_

### Official Review · Reviewer_tUxi · 2024-10-27

**Soundness:** 3
**Presentation:** 3
**Contribution:** 3
**Rating:** 6
**Confidence:** 3

**Summary:**

The submission presents a method to improve conformal training (Stutz et al 2021) for classification through an adapter-based tuning method. The authors claim significant efficiency improvements. The authors draw attention to an existing with conformal training - specifically that the regularization term in conformal training may deteriorate classifier accuracy, and offer a method to alleviate this. The key idea is to use intra-order-preserving functions (Rahimi et al. 2020) as an extra component applied to a pre-trained model. The authors demonstrate the synthesis of two methods through CIFAR-100 and ImageNet classification tasks.

**Strengths:**

To the best of my knowledge, no other work has introduced an “adapter” for conformal training to improve the efficiency of prediction sets. It combines two interesting works and presents interesting empirical results, albeit in limited settings. And could be interesting to the community. Given the clarifications in the authors’ response, I would be willing to increase the score.

The submission is clear, mostly technically correct, and experimentally rigorous. The main strength of the paper is its empirical findings. Although only applied to limited settings/tasks, the empirical results support their claims. Evaluating the effectiveness of C-Adapter on other tasks (NLP or time series), more SOTA architectures and potentially more impactful applications would provide a more comprehensive understanding of its capabilities and limitations.

**Weaknesses:**

The main weakness of the submission is that it lacks theoretical insight into the efficacy of the method. When does C-adapter perform better/worse than conformal training? Is this always guaranteed? Furthermore, the submission could use an in-depth theoretical analysis of the robustness of C-Adapter - its robustness to distribution shifts, adversarial examples, and noisy data. Theoretical investigations would significantly strengthen the claims of robustness made based on empirical observations.

**Questions:**

With that being said, I have several concerns and questions for the authors:
1. The main motivation behind his paper is that increasing \lambda decreases the classification accuracy, leading to larger average size of prediction sets. However, this is not the case in Fig 1 (blue lines) presented - CIFAR-100 has a U shape, and ImageNet is almost flat. Please explain why this could be the case and why the relationships could differ between datasets.
2. The authors present a training objective designed to optimize the efficiency of conformal predictors over the entire range of alpha values (0, 1). Please explain why this was done. Practically, maybe only one \alpha value could be used. Does only optimizing for one \alpha value improve performance compared to all?
3. The authors test C-Adapter's performance when trained on ImageNet and then tested on ImageNet-V2. This approach evaluates how well the adapted model generalizes to a different but related dataset, simulating a distribution shift scenario. However, the authors say in page 10 - “Notably, coverage will not be affected under this setting, as the calibration and test sets remain exchangeable”. A distribution shift means that calibration and test sets are non-exchangeable. If this is the case, the claim that C-adapter is robust to distribution shift is not warranted. Please explain. Furthermore, quantification of the distribution shift (if there is) would significantly strengthen this claim.
4. Will the code and data be publicly available?

---

> ### Author Response · Authors · 2024-11-20
> **Response to Reviewer tUxi (1/2)**
>
> Thank you for the valuable comments and detailed feedback on our manuscript. Please find our response below.
>
> **1. Broader application of C-Adapter**
>
> Thank you for the suggestion. We extended our experiments to the task of **text classification** using **large language models**. We present the experimental setting and results in Appendix I. In particular, we adopt conformal prediction in the 14-class classification of dbpedia 14 using LLama3-8B. The results in **Table 7** of Appendix I show that C-Adapter can work well in this application. We also put our results here for your reference. The results are organized by (w/o C-Adapter) / (w/ C-Adapter).
>
> | Score | Coverage ($\alpha = 0.05$) | Size ($\downarrow$) ($\alpha = 0.05$) | Coverage ($\alpha = 0.1$) | Size ($\downarrow$) ($\alpha = 0.1$) |
> |-------|----------------------------|---------------------------------------|---------------------------|--------------------------------------|
> | THR   | 0.94 / 0.95                | 2.80 / **2.61**                      | 0.89 / 0.89               | 2.17 / **2.04**                     |
> | APS   | 0.95 / 0.94                | 3.14 / **2.75**                      | 0.90 / 0.91               | 2.33 / **2.08**                     |
> | RAPS  | 0.95 / 0.95                | 3.23 / **3.11**                      | 0.90 / 0.90               | 2.48 / **2.32**                     |
> | **Average** | -                      | 3.06 / **2.82**                      | -                         | 2.33 / **2.15**                     |
>
>
>
> In addition, we'd like to emphasize the model-agnostic advantage of C-Adapter in real applications. For example, C-Adapter can be applied to CLIP models (as shown in Table 1), which are multi-modal vision and language models. C-Adapter requires access only to the model outputs and integrates effortlessly with any classifier (even black-box models), regardless of the network architecture or pre-training strategy.
>
> **2. Theoretical insight of C-Adapter? [W1]**
>
> First, we clarify that our method is orthogonal to ConfTr and can be used to improve ConfTr. In particular, we can append the C-Adapter to models trained by ConfTr and further enhance the performance of conformal prediction (See Figure 4). Therefore, the contribution of this work does not necessarily depend on the simple comparison between C-Adapter and ConfTr.
>
> **Cost of accuracy.** As presented in the motivation (Figure 1 and Figure 11), the performance of ConfTr is limited because its regularization inevitably deteriorates the classifier accuracy. While ConfTr may improve the efficiency with a specific $\alpha$, **the cost of accuracy is generally unacceptable** and potentially limits the improvements. In the revised paper, **we provide a theoretical analysis in Appendix K to show the effect of topK accuracy on the bounds of the expected set size**: the lower bound of the expected set size is negatively related to the top-k accuracy. Therefore, the cost of accuracy introduced by ConfTr will increase the lower bound of the expected size, leading to suboptimal performance in efficiency. This highlights the importance of preserving top-k accuracy, which establishes the advantage of our method. Notably, our method can adapt pre-trained classifiers for conformal prediction while preserving the top-k accuracy of the classifiers unchanged, rather than mitigating accuracy drops (as demonstrated in Theorem 1).
>
> **Overall efficiency.** Our loss function is novel and is superior to the Size loss of ConfTr (See Table 2). While ConfTr only optimizes the efficiency at a pre-defined error rate $\alpha$, they cannot ensure the optimization of the overall efficiency in Eq.(4), which limits the application of ConfTr after training. Differently, our method **does not require to define a specific $\alpha$** during training and directly optimize the discriminability of non-conformity scores between correctly and randomly matched data-label pairs. **Through Proposition 1, we show that optimizing our loss function is theoretically equivalent to minimizing the overall efficiency**.
>
> **Flexibility.** C-Adapter requires access only to the model outputs and seamlessly integrates with any classifier, including black-box models. In contrast, ConfTr necessitates modifying model weights, which limits its applicability.
>
> In summary, we provide theoretical analysis (Propositions 1, 2, 3 and Theorem 1) to demonstrate the advantages of our method in accuracy preserving and overall efficiency, respectively. Moreover, we'd like to clarify that our work is not a simple combination of two previous works. In this work, we implement the adapter layer for accuracy preserving and proposes a novel loss function for its optimization, which establish the new SOTA training method for conformal prediction.

---

> ### Author Response · Authors · 2024-11-20
> **Response to Reviewer tUxi (2/2)**
>
> **3. Clarification of distribution shifts [W2, Q3]**
>
> There might be some misunderstandings. We clarify that the distribution shifts we mentioned happen between the training set and calibration/test sets. Thus, coverage is not affected under this setting, as the calibration and test sets is sampled from the same dataset (e.g., ImageNet-V2). We present this analysis to show that C-Adapter can work well even though it is trained on a different distribution from the calibration/test set. Specifically, C-Adapter is tuned to enhance the efficiency of conformal predictors on ImageNet, while still improving efficiency on ImageNetV2.  In the revised paper, we rewrite this paragraph to avoid any misunderstandings.
>
> **4. Clarification of Figure 1 [Q1]**
>
> Thank you for pointing out the ambiguous description. We clarify that the decreasing of classifier accuracy would **limits the improvement of conformal prediction** from ConfTr. We have corrected the description in the revised paper. In Figure 1a, we show that the improvement of ConfTr is reduced after the accuracy is decreased with a large $\lambda$ (for THR, ConfTr consistently has a negative effect due to the accuracy drop). In Figure 1b, we show that ConfTr cannot improve the efficiency of prediction sets on ImageNet with different $\lambda$. The results consistently demonstrate the suboptimal performance of ConfTr on different scales of application. This motivates our approach to adapting pre-trained classifiers for conformal prediction without compromising accuracy.
>
> As for the difference of performance between the datasets, we conjecture that it might be related to the class numbers of datasets: ImageNet contains 1000 classes, leading to a large Size Loss and making it more challenging to balance the classification loss with Size Loss. A larger $\lambda$ causes a significant drop in model accuracy, whereas a smaller $\lambda$ fails to enhance efficiency. Consequently, applying ConfTr to large-scale datasets such as ImageNet becomes more challenging.
>
>
> **5. Clarification of the proposed loss function [Q2]**
>
> Yes, this is exactly the advantage of our proposed loss function over the Size loss of ConfTr. In particular, ConfTr optimizes the average set size at a pre-defined error rate $\alpha$ but hope it can generalize to all error rates. Instead, **our loss function is not designed for a specific error rate**. To optimize the overall efficiency defined in Eq.(4), we translate it into an equivalent form as shown in Eq.(5). In proposition 1, we formally prove that minimizing the probability in Eq.(5) is equivalent to optimizing the overall efficiency defined in Eq.(4). Intuitively, our goal is to enhance the discriminability of non-conformity scores between correctly and randomly matched data-label pairs, which translates to more efficient conformal prediction sets at various coverage rates. We provide an illustration of score distribution in Figure 3 and validate that C-Adapter promotes highly distinguishable scores between correct and incorrect labels. In addition, we also provides an **ablation study of loss functions in Table 2** and we implement the size loss of ConfTr in C-Adapter for comparison. The results show that **the size loss (only optimizing for one \alpha value) is inferior to our proposed loss** in the performance of conformal prediction.
>
>
> **6. Will the code and data be publicly available? [Q4]**
>
> Yes, we will definitely make our code and data publicly available once the paper is published.

---

> ### Comment · Reviewer_tUxi · 2024-11-25
>
> Thank you for your thorough response and updated manuscript. Based on your clarifications and comments, I have decided to raise your score.

---

> > ### Author Response · Authors · 2024-11-25
> >
> > Thank you for reviewing our response and increasing the score. We are delighted that our response addressed your concerns. Your feedback is highly valuable in improving the quality of this work.

---

### Official Review · Reviewer_8WmQ · 2024-10-30

**Soundness:** 3
**Presentation:** 3
**Contribution:** 2
**Rating:** 6
**Confidence:** 3

**Summary:**

The authors proposed an adapter-based tuning strategy to enhance conformal prediction performance without sacrificing the model's performance. They implemented this adapter as a class of intra-order-preserving functions to maximize the discriminability of non-conformity scores between correctly and randomly matched data-label pairs. This approach achieved high non-conformity scores for incorrect labels, enhancing the efficiency of prediction sets across different coverage rates.

**Strengths:**

**S1.** The paper is very well written and presented.

**S2.** The methodological development is well written, with a comprehensive background and related works.

**S3.** I found the experimentation well-motivated, covering important ablation studies, including alpha, training strategy, adaptation strategy, and parameter $T$. The authors achieved considerable improvement across different feature extractor backbones. Besides, the experiments on alpha under THR and APS demonstrate the robustness of their C-Adapter strategy.

**Weaknesses:**

**W1. Methodological Novelty.** I found the contributions made by the authors are somewhat limited. The intra order-preserving function is adapted from (Rahimi et al., 2020). Overall, the complete approach is somewhat like a combination of the existing SOTAs, including the intra order-preserving function, conformal training (Rahimi et al., 2020, Stutz et al., 2021). Other than the theoretical demonstration and an additional learnable layer (the adapter layer), I would suggest the authors to specifically highlight any other methodological contribution.

**W2. Loss function explanation.** One of the limitations of their work is the quality of the approximation depends heavily on the choice of $T$, which seems to be not affected by the prediction set size that much, according to the experimental findings. What is the rationale behind the insensitiveness of their approach to this parameter $T$? Besides, they approximated the loss function with sigmoid which might not be strictly convex over the entire domain. Hence, this approximation might introduce non-convexities that could impact the convergence and optimization process.

While there are some concerns about the proposed methodology and the extent of its novelty, the current rating reflects the accumulated efforts in problem definition, motivation, presentation, diverse experimentation, and ablation studies. I would strongly suggest to consider the findings and address them in their rebuttal. Good luck.

**Questions:**

I tried to cover most of my concerns and questions in the *Weaknesses* section. I kindly request the authors to review that section.

---

> ### Author Response · Authors · 2024-11-20
> **Response to Reviewer 8WmQ**
>
> Thank you for your positive feedback and valuable comments on our manuscript. Please find our response below.
>
> **1. Clarification of methodological novelty [W1]**
>
> There might be some misunderstandings. We clarify that our loss function in Eq.(7) is novel as one of main contributions in this work (See the 2nd contribution in Introduction). It is totally different from the Size loss used in ConfTr. While the Size loss in ConfTr minimizes the prediction sets at a pre-defined error rate (e.g., $\alpha=0.01$), our loss function is design to separate the  scores of correctly and incorrectly matched data-label pairs, which is theoretically equivalent to optimizing the overall efficiency in Eq.(4) (See Proposition 1). Thus, the methodological novelty of this work is sufficient with the accuracy-preserved design for conformal prediction and the new loss function, as appreciated by reviewer itRT.
>
> In the revised paper, we also provide a new theoretical analysis in **Appendix K** to show the effect of accuracy cost on the efficiency of prediction sets. The analysis formally shows that the cost of accuracy introduced by ConfTr will increase the lower bound of the expected size, leading to suboptimal performance in efficiency. The theoretical results provide deep insights for designing effective training methods in conformal prediction.
>
>
> **2. Why C-Adapter is insensitive to $T$ [W2]:**
>
> Sorry for the confusion. We clarify that the insensitivity is due to the fact that the value of $T$ (in the range of $[10^{-6}, 10^{-2}]$) is sufficiently small to ensure a high-quality approximation. The previous analysis of $T$ presented in Figure 7 is limited to the the range $[10^{-6}, 10^{-2}]$, which are too small to show the effect of $T$. To avoid any misunderstanding, we extend the range of $T$ to $[10^{-6}, 10^1]$ in the revised version. As presented in Figure 7, a large $T$ (e.g., 1, 10) leads to poor performance in the efficiency of prediction sets. With the decrease of $T$, the performance of C-Adapter is improved because the sigmoid function gradually approximates the indicator function. Note that the performance is converged when $T$ is sufficiently small (e.g., 0.01), our method **does not require a heavy tuning** for $T$ as we can simply set a small $T$.
>
>
>
> **3. Clarification of using Sigmoid as the surrogate function [W2]**
>
> Yes, using the sigmoid function cannot remain strictly convex over the entire domain.  Yet, we argue that the sigmoid function is still commonly used in deep learning due to its non-linearity and differentiability. For example, it is generally adopted as *the activation function in deep neural networks* [1], the "switch" of neurons, and the gating mechanism controlling information flow in *recurrent neural networks* (RNNs) and *long short-term memory networks* (LSTMs) [2]. Notably, Sigmoid is also **common used** as the surrogate functions for the indicator function in *AUC maximization* [3,4].
>
> As for the convergence, we provide a empirical analysis in **Appendix F** to show the **fast convergence of C-Adapter**, approaching nearly optimal performance in just 50 iterations. This advantage of C-Adapter may arise from the intra order-preserving functions, which significantly reduce the hypothesis space in learning.
>
> Furthermore, we provide an empirical comparison to show the advantage of Sigmoid in **Table 8** (Appendix I of the revised paper). We compare the performance of the most commonly used surrogates for the indicator function [4]: **Square**, **Hinge**, and **Sigmoid**. The results show that **Sigmoid** consistently outperforms the other two functions across all non-conformity scores.
>
> We also present the results here for your reference:
> |  | Baseline | Hinge | Square | Sigmoid |
> |-----------------|----------|-------|--------|---------|
> | **THR**         | 5.66     | 5.51  | 5.47   | **5.41** |
> | **APS**         | 20.00    | 5.91  | 5.88   | **5.73** |
> | **RAPS**        | 10.28    | 7.49  | 7.35   | **6.53** |
> | **Average**     | 11.98    | 6.30  | 6.23   | **5.89** |
>
> The experiment is conducted on ImageNet with DenseNet121. The error rate $\alpha$ is set to 0.05. Since all methods achieve the desired coverage, only Size is reported here.
>
> [1] Sharma, Sagar, Simone Sharma, and Anidhya Athaiya. "Activation functions in neural networks." Towards Data Sci 6.12 (2017): 310-316.
>
> [2] Sherstinsky, Alex. "Fundamentals of recurrent neural network (RNN) and long short-term memory (LSTM) network." Physica D: Nonlinear Phenomena 404 (2020): 132306.
>
> [3] Yan, Lian, et al. "Optimizing classifier performance via an approximation to the Wilcoxon-Mann-Whitney statistic." Proceedings of the 20th international conference on machine learning (icml-03). 2003.
>
> [4] Yang, Tianbao, and Yiming Ying. "AUC maximization in the era of big data and AI: A survey." ACM Computing Surveys 55.8 (2022): 1-37.

---

### Official Review · Reviewer_itRT · 2024-11-01

**Soundness:** 4
**Presentation:** 3
**Contribution:** 4
**Rating:** 8
**Confidence:** 3

**Summary:**

The paper "C-Adapter: Adapting Deep Classifiers for Efficient Conformal Prediction Sets" introduces C-Adapter, a method that improves the efficiency of conformal predictors while preserving classification accuracy. By adding an adapter layer to trained classifiers, C-Adapter maintains top-k accuracy through label ranking preservation. It optimizes a unique loss function to enhance non-conformity score separation between correct and incorrect predictions, resulting in more efficient prediction sets. Tested on CIFAR-100 and ImageNet, C-Adapter significantly reduces prediction set sizes and outperforms existing methods like Conformal Training, adapting well across various classifiers and scoring functions with minimal computational cost.

**Strengths:**

Originality: The paper is original in proposing C-Adapter, an adapter-based method for improving the efficiency of conformal predictors while maintaining accuracy. Unlike traditional methods such as Conformal Training, which can compromise classifier performance, C-Adapter innovatively integrates an adapter layer that preserves label ranking to maintain top-k accuracy. The introduction of intra order-preserving functions and a new loss function tailored for conformal prediction is novel and adds depth to the methodology.

Quality: The quality of the work is strong, supported by both theoretical justifications and comprehensive empirical results. The authors provide a solid mathematical foundation for their approach, including proofs and detailed discussions on the properties of the proposed method. The experiments are well-designed and conducted across various benchmarks, such as CIFAR-100 and ImageNet, using multiple classifiers. This extensive evaluation highlights the robustness and effectiveness of C-Adapter. The paper also compares its method with existing solutions like Conformal Training and demonstrates clear improvements.

Clarity: The paper is generally clear, with a well-organized structure that guides the reader through the problem, methodology, and experimental results. The introduction and related work sections set the stage effectively, and the results are presented with informative figures and tables. However, the clarity could be further enhanced by simplifying some complex mathematical sections and providing more intuitive explanations. This would make the paper more accessible to readers who are not specialists in conformal prediction or the specific mathematical frameworks used.

Significance: The paper's contribution is significant, particularly for the field of uncertainty quantification in machine learning. C-Adapter presents a practical and adaptable solution that can be applied to a variety of classifiers and settings, including black-box models. Its ability to maintain classification accuracy while reducing prediction set sizes has practical implications for high-stakes applications such as medical diagnostics and financial forecasting, where efficient and reliable uncertainty estimates are crucial. The method's flexibility and minimal computational overhead further enhance its significance, positioning it as a valuable tool for both research and practical implementations.

**Weaknesses:**

While the paper provides strong theoretical support, certain sections, particularly those involving the mathematical underpinnings of intra order-preserving functions, may be difficult for readers unfamiliar with this concept. To improve accessibility, the authors could include a simplified overview or illustrative examples to help readers intuitively grasp the key ideas without needing extensive background knowledge. This would broaden the paper’s reach and make it more appealing to a wider audience.

While the paper briefly addresses distribution shifts using ImageNet-V2, a more detailed exploration or comparison with other methods in this context would strengthen the claim of C-Adapter’s robustness. Further experiments with synthetic or real-world data shifts could provide deeper insights into its performance under more varied conditions.

While the paper claims C-Adapter is insensitive to hyperparameters, the provided analysis on the parameter T is limited. A more comprehensive exploration of hyperparameter sensitivity, including the impact of different tuning strategies and settings, would help verify this claim. Showing how C-Adapter behaves under a variety of hyperparameter configurations can reassure practitioners of its reliability in different scenarios.

**Questions:**

While C-Adapter is shown to outperform Conformal Training (ConfTr), what are the specific conditions or datasets where ConfTr might still be preferable or complementary to C-Adapter?

The results indicate that C-Adapter improves conditional coverage. What are the underlying mechanisms that enable this improvement and how it compares to methods specifically tailored for conditional coverage?

The evaluation focuses on standard score functions (THR, APS, RAPS). How would C-Adapter perform with more specialized or non-standard score functions used in specific domains?

---

> ### Author Response · Authors · 2024-11-20
> **Response to Reviewer itRT**
>
> Thank you for your positive feedback and valuable comments. Please find our response below.
>
> **1. Simplified overview of Intra Order-Preserving Function [W1]**
>
> Thank you for the suggestion. The core idea of intra order-preserving function we implemented is to decouple the label ranking and the logit values in the tuning. In particular, we begin by preserving a duplicate of the label ranking, and then transmit the logit values to the linear layer for processing. Finally, we recover the label ranking in the final output, keeping the order unchanged. In the revised manuscript, we present this core idea in the section 3 and offer a detailed explanation (includes an illustration) of intra order-preserving functions in Appendix D.
>
> **2. More results of C-Adapter under data shifts [W2]**
>
> Thank you for the suggestion. In the revised version, we provide additional results on *ImageNet-A* (Hendrycks et al., 2021a) and *ImageNet-R* (Hendrycks et al., 2021b). In particular, ImageNet-A focusing on adversarial examples that are modified to mislead models, and ImageNet-R consisting of images transformed by various artistic styles and visual changes to test models' adaptability to different visual distributions. We present the detailed results of these two benchmarks in Table 6 of **Appendix I**. Specifically, C-Adapter can significantly improve the performance of popular non-conformity scores in both the two datasets. The reults further confirms the effectiveness of our methods in the scenarios of distribution shifts between the training set and test/calibration set.
>
> **3. Clarification of C-Adapter's sensitivity to $T$ [W3]**
>
> To extensively analyze the effect of $T$, we extend the range of $T$ to $[10^{-6}, 10]$ in the hyparameter sensitivity analysis. In **Figure 7** of the revised manuscript, We show that C-Adapter achieves better performance with a smaller value of $T$. And, C-Adapter with a sufficiently small $T$ (e.g., 0.01) can effectively improve the effciency of conformal prediction. It is because Sigmoid function in Eq.(7) can approximate the indicator function with a small $T$. Thus, our method does not require heavy tuning of hyperparameter, as we can simply set a small $T$.
>
> **4. Clarification of relations between C-Adapter and ConfTr [Q1]**
>
> We clarify that our method is complementary to ConfTr, as C-Adapter can be implemented after training with ConfTr. The results in Figure 4 show that our method can outperform and improve ConfTr. As for the potential benefits of ConfTr to our method, current results show that C-Adapter+ConfTr does not outperform applying C-Adapter alone in all cases. It might be because the regularization term of ConfTr normally damages the classification accuracy, leading to suboptimal performance in conformal prediction. In future work, training methods may benefits C-Adapter if a new training objective is proposed to improve both accuracy and conformal prediction.
>
> **5. How C-Adapter benefits the conditional coverage [Q2]**
>
> Thank you for the suggestion. In Appendix J, we add a gradient analysis of the proposed loss function to explain the benefits of C-Adapter. The challenge of poor conditional coverage metrics typically arises from the performance variation across sub-groups of data, leading to disparities in score distributions among these groups. C-Adapter mitigates the discrepancies through optimizing the proposed loss function in Eq.(8). In particular, the gradient analysis shows that our loss function enables to put more focus on samples with high scores, decreasing the variation in non-conformity scores among data samples. In this way, our method can improve the conditional coverage with consistent performance across different data sub-groups.
>
> As for previous methods to improve the conditional coverage, they normally alleviate this issue by computing thresholds for different sub-groups. For example, Clustered Conformal Prediction (CCP) [1] clusters the classes based on their similarities and calculates group-specific thresholds to perform conformal prediction. While CCP can benefits the conditional coverage, they usually leads to larger prediction sets than vanilla conformla prediction (See Table 2 in their paper [1]). In addition, the improvement of CCP relies heavily on the quality of clustering, which requires a large caibration set. This highlights the advantage of our method, which can improve both the conditional coverage and the average set size.
>
> [1] Ding, Tiffany, et al. "Class-conditional conformal prediction with many classes." Advances in Neural Information Processing Systems 36 (2024).
>
>
> **6. Evalutions on non-standard score functions [Q3]**
>
> Thank you for the suggestion. We are more than willing to provide extra results with non-standard score functions. We will highly appreciate it if you could provide an example of such score functions or related works.

---

### Official Review · Reviewer_FZnX · 2024-11-03

**Soundness:** 3
**Presentation:** 3
**Contribution:** 3
**Rating:** 8
**Confidence:** 3

**Summary:**

The paper introduces C-Adapter, an adapter-based tuning method designed to improve the efficiency of conformal predictors without compromising classification accuracy. This approach is highly relevant to uncertainty quantification, where conformal prediction frameworks generate prediction sets that, with a specified coverage rate, are likely to include the true class. C-Adapter seeks to optimize prediction efficiency while preserving or enhancing model accuracy, which holds significant potential for high-stakes applications such as medical diagnostics.

**Strengths:**

1. The results demonstrate that the proposed method significantly reduces prediction set sizes while maintaining accuracy.
2. C-Adapter is versatile, working effectively with a range of classifiers and showing strong compatibility with black-box models.
3. Empirical results indicate that C-Adapter performs consistently across various datasets, models, and evaluation metrics.
4. Minimal hyperparameter tuning and high computational efficiency make C-Adapter highly practical for deployment.

**Weaknesses:**

1. The primary concern with this paper is the lack of comparison with related methods. The authors tested the proposed C-Adapter across various benchmarks (Table 1), loss functions (Table 2), values of α (Table 3), and distribution shifts (Table 4), but did not include comparisons with other approaches in conformal prediction.
2. While the use of adapters for conformal prediction is a novel application, the concept of adapters itself is well-established. The insight for choosing adapters over other modules, such as LoRA, is not sufficiently discussed and would benefit from further elaboration.

**Questions:**

1. The authors are encouraged to include some related methods in the comparisons to provide a more comprehensive evaluation.
2. The motivation and insight for employing the Adapter module should be further emphasized to clarify its significance in the proposed approach.

---

> ### Author Response · Authors · 2024-11-20
> **Response to Reviewer FZnX**
>
> Thank you for the valuable comments and detailed feedback on our manuscript. Please find our response below.
>
> **1. Comparison with related methods [W1 and Q1]**
>
> Thank you for the suggestion. However, we'd like to clarify that our method is **orthogonal** to current methods of conformal prediction, as the first adapter-based method in this area. Therefore, we provide extensive experiments to show that our method can enhance the performance of existing methods, including non-conformity scores -- THR, APS, and RAPS (see Table 1) and training algorithm -- ConfTr (see Figure 4). We are willing to supplement more results, if you can explicitly provide some approaches we missed in the paper.
>
> **2. Why employing the adapter module [W2 and Q2]**
>
> Thank you for the suggestion. In the revised version, we update the related work (Appendix A) to emphasize the distinctions of our method and adapters in other tasks. In the literature, adapters are generally designed as an efficient method to adapt pretrained models for downstreaming tasks [1-5], which plays a similar role as LORA in LLM. While our method shares the same concept of adapter, its underlying insight is totally different from previous adapters.
>
> As the training objective of conformal prediction may deteriorate the accuracy, we hope to **preserve the label ranking** in the model output. Therefore, our C-Adapter only appends an adapter layer to the output layer of original models, enabling the implementation of *intra order-preserving functions*. Differently, previous adapters generally insert adapter layers between the existing layers of a neural network. Therefore, they cannot preserve the label ranking (as well as other PEFT methods, like LORA), making it suboptimal for conformal prediction. In the ablation study (Page 8), we empirically show that C-Adapter outperforms other fine-tuning methods, including retraining and linear probing.
>
> In summary, we employ C-Adapter to *enable the efficient adaptation of trained classifiers for conformal prediction without sacrificing classification accuracy* (line 45). Compared to ConfTr and other PEFT methods, our method does not required to update the parameters of original models (**high efficiency**) and maintain the classification accuracy (**more effective**).
>
>
>
> [1] Sylvestre-Alvise Rebuffi, Hakan Bilen, and Andrea Vedaldi. Learning multiple visual domains with residual adapters. arXiv:1705.08045 [cs, stat], November 2017.
>
> [2] Houlsby, Neil, et al. "Parameter-efficient transfer learning for NLP." International conference on machine learning. PMLR, 2019.
>
> [3] Zhaojiang Lin, Andrea Madotto, and Pascale Fung. Exploring versatile generative language model via parameter-efficient transfer learning. In Findings of the Association for Computational Linguistics: EMNLP 2020, pp. 441–459, Online, November 2020. Association for Computational Linguistics. doi: 10.18653/v1/2020.findings-emnlp.41.
>
> [4] Hu, Edward J., et al. "Lora: Low-rank adaptation of large language models." arXiv preprint arXiv:2106.09685 (2021).
>
> [5] Sung, Yi-Lin, Jaemin Cho, and Mohit Bansal. "Vl-adapter: Parameter-efficient transfer learning for vision-and-language tasks." Proceedings of the IEEE/CVF conference on computer vision and pattern recognition. 2022.

---

> > ### Comment · Reviewer_FZnX · 2024-11-25
> >
> > This reviewer appreciates the authors’ responses to my concerns. All of my questions have been addressed, and I have decided to adjust my original score to “accept.”

---

> > > ### Author Response · Authors · 2024-11-25
> > >
> > > Thank you for reviewing our response and raising your score. We are pleased that our response addressed your concerns, which also improves the quality of this work.

---

### Author Response · Authors · 2024-11-20
**General Response**

We sincerely thank all the reviewers for their time, insightful suggestions, and valuable feedback. We are pleased that the reviewers recognize the **novelty** of this work (itRT, tUxi), and point out that it will be **interesting** to the community (tUxi) with **significant contribution** (itRT). The reviewers appreciate that C-Adapter is **versatile, robust, and effective** (FZnX, itRT, 8WmQ), as a **practical and adaptable** solution (FZnX, itRT). Besides, we are also encouraged that reviewers find the empirical results are **comprehensive, rigorous, well-designed and motivated** (itRT, 8WmQ, tUxi) with **clear and significant** improvements (FZnX, itRT, 8WmQ). Reviewers recognize that the theoretical results are **solid** (itRT) and the writing is **clear, well-organized, technically correct**, with **comprehensive** background and related works (itRT, 8WmQ, tUxi)

In the following responses, we have addressed the reviewers' comments and concerns point by point. The reviews allow us to strengthen our manuscript and the changes$^1$ are summarized below:
* Added related works to discuss different adapters in **Appendix A**. [FZnX]
* Added overview and details for intra order-preserving function in **Line 199-202** and **Appendix D**. [itRT]
* Added experiments for robustness on more benchmarks in **Appendix I**. [itRT]
* Revised hyperpramater analysis in **Figure 7** and **Line 482-485**. [itRT, 8WmQ]
* Added explanation on conditional coverage in **Appendix J**. [itRT]
* Added experiments on the Sigmoid Function in **Appendix I**. [8WmQ]
* Added theoretical analysis on the effect of top-k accuracy on conformal prediction in **Appendix K** [tUxi]
* Clarified description on the performance of ConfTr in the caption of **Figure 1** [tUxi]
* Clarified description on the robustness to distribution shifts in **Line 500-502** [tUxi]
* Added experiments on text classification using LLMs in **Appendix I**. [tUxi]

$^1$ For clarity, we highlight the revised part of the manuscript in **blue** color.

---

### Comment · Area_Chair_c1X3 · 2024-11-25
**Please engage in the discussion**

Dear all,

Many thanks to the reviewers for their constructive reviews and the authors for their detailed responses.

Please use the next ~2 days to discuss any remaining queries as the discussion period is about to close.

Thank you.

Regards,

AC

---

### Note · Authors · 2025-03-19

**Comment:**

I have read and agree with the venue's withdrawal policy on behalf of myself and my co-authors.

During the preparation for open-sourcing the code, we conducted a comprehensive review of the implementation and identified certain issues that impact some of the reported results. These issues may have led to over-claimed conclusions, and we believe it is essential to address them thoroughly to ensure the integrity and accuracy of our work. As such, we have decided to withdraw the current submission and resolve these issues.

We sincerely apologize for any inconvenience this may cause and deeply appreciate the time and effort the committee and reviewers have invested in evaluating our work.

**Withdrawal Confirmation:**

I have read and agree with the venue's withdrawal policy on behalf of myself and my co-authors.

---

### Meta-Review · Area_Chair_c1X3 · 2024-12-19

**Metareview:**

The paper introduces the Conformal Adapter (C-Adapter), a new method designed to improve the efficiency of conformal prediction sets while maintaining classification accuracy. This post-processing layer retains label ranking in output logits and optimises conformal prediction efficiency. It outperfroms existing methods like Conformal Training (ConfTr) in efficiency and robustness across various datasets and models.

The paper has several contributions, summarised as follows:
1) Utilises an "intra order-preserving function" to retain classification accuracy and introduces a loss function to enhance data-label pair discriminability.
2) Demonstrates significant improvements in prediction set efficiency across benchmarks such as CIFAR-100, ImageNet, and ImageNet-V2, and is compatible with multiple score functions and model architectures.
3) Requires minimal hyperparameter tuning, is computationally efficient, and easy to integrate with existing models, including black-box models.
4) Provides detailed experiments and analyses, with a strong theoretical foundation connecting the loss function to efficiency optimisation.

Overall, the paper makes a good contribution to conformal prediction literature by offering a practical method for enhancing prediction set efficiency, suggesting future research might explore broader aspects of conformal prediction.

**Additional Comments On Reviewer Discussion:**

Reviewers have been very supportive of this work, as evidenced by the scores. It is also worth highlighting that the authors did a very good rebuttal that included a paper revision too. Given the paper's contributions, clarity, originality, and novelty this could be accepted as a spotlight.

---

### Decision · Program_Chairs · 2025-01-22

Accept (Poster)